# MYOD-SKP2 axis boosts tumorigenesis in fusion negative rhabdomyosarcoma by preventing differentiation through p57$^{Kip2}$ targeting

Rhabdomyosarcomas (RMS) are pediatric mesenchymal-derived malignancies encompassing PAX3/7-FOXO1 Fusion Positive (FP)-RMS, and Fusion Negative (FN)-RMS with frequent RAS pathway mutations. RMS express the master myogenic transcription factor MYOD that, whilst essential for survival, cannot support differentiation. Here we discover SKP2, an oncogenic E3-ubiquitin ligase, as a critical pro-tumorigenic driver in FN-RMS. We show that SKP2 is overexpressed in RMS through the binding of MYOD to an intronic enhancer. SKP2 in FN-RMS promotes cell cycle progression and prevents differentiation by directly targeting p27$^{Kip1}$ and p57$^{Kip2}$, respectively. SKP2 depletion unlocks a partly MYOD-dependent myogenic transcriptional program and strongly affects stemness and tumorigenic features and prevents in vivo tumor growth. These effects are mirrored by the investigational NEDDylation inhibitor MLN4924. Results demonstrate a crucial crosstalk between transcriptional and post-translational mechanisms through the MYOD-SKP2 axis that contributes to tumorigenesis in FN-RMS. Finally, NEDDylation inhibition is identified as a potential therapeutic vulnerability in FN-RMS.

Rhabdomyosarcoma (RMS) is a myogenic malignancy that accounts for 50% of pediatric soft tissue sarcomas and 7% of all pediatric cancers[1]. The two major histological RMS subtypes, alveolar and embryonal, have recently been molecularly reclassified as Fusion Positive (FP-RMS) and Fusion Negative (FN-RMS), depending on the presence or absence of PAX3/7-FOXO1 gene fusions[2,3]. PAX3-FOXO1 FP-RMS mainly relies on the oncogenic fusion protein for survival, while FN-RMS shows chromosomal aberrations and mutations of the RAS pathway[4]. Both subtypes, as pediatric cancers, have low mutational burdens suggesting epigenetic mechanisms may regulate tumorigenesis[5–7]. Despite aggressive multimodality therapeutic regimens, which can result in life-altering side effects, the 3-year overall survival for patients affected by high-risk forms, irrespective of gene fusion, remains less than 30%[1].

RMS cells are characterized by sustained expression of the master Myogenic Regulatory transcription Factors (MRFs) MYOD and MYOG, yet have lost the ability to differentiate into skeletal muscle and proliferate indefinitely[8]. In this scenario, unexpectedly, MYOD appears fundamental for the survival of RMS cells rather than able in inducing the myogenic process, with mechanisms still to be fully elucidated[6,7].

Tightly controlled regulation of the proteasomal degradation of key proteins represents one of the major mechanisms controlling normal myogenic differentiation[9–11].

The S-phase kinase-associated protein-2, SKP2, is an F-box protein and a critical substrate recognition subunit of the E3 ubiquitin ligase SKP1-Cullin1-F-box protein (SCF) complex, also named Cullin-Ring Ligase 1 (CRL1). SKP2-SCF/CRL1 complex ubiquitylates Cyclin Dependent Kinase (CDK) inhibitors (CDKI) such as p21$^{Cip1}$ and p27$^{Kip1}$,

e-mail: rossella.rota@opbg.net

targeting them for degradation, thus promoting cell cycle progression[12]. Several lines of evidence show that SKP2 behaves as an oncogene[13–15]. Indeed, SKP2 expression is often deregulated in cancer through gene amplification or overexpression and is mainly linked to tumor cell proliferation and stemness. Consistently, SKP2 targeting causes cell cycle arrest of cancer cells and halts tumorigenesis in vivo[16–18].

Here, we identify a MYOD-SKP2 axis that governs differentiation and boosts tumorigenesis in FN-RMS. We show that *SKP2* is highly expressed in RMS and its expression is regulated by MYOD through the binding to an intronic enhancer region within the *SKP2* gene locus in loop with its promoter. SKP2 has several pro-tumorigenic functions in FN-RMS promoting cell cycle progression and blocking terminal differentiation by targeting p27[Kip1] and p57[Kip2], respectively. Moreover, SKP2 hampers the transcriptional activity of MYOD on myogenic genes and impairs p57[Kip2]-dependent stabilization of MYOD. SKP2 genetic depletion arrests cell proliferation, inhibits stemness and restores myogenic differentiation in vitro and in vivo preventing tumor growth. SKP2 inhibition with a small molecule, SMIP004, mirrors the in vitro effects of silencing. We also show that the NEDDylation inhibitor MLN4924 mirrors the effects of SKP2 depletion and suppresses FN-RMS growth in vitro and in vivo. Collectively, these results shed light on the interconnections between transcriptional and post-translational networks driving FN-RMS pathogenesis, unveil unknown MYOD and SKP2 functions and suggest a therapeutic approach to treat FN-RMS.

## Results

### SKP2 is up-regulated in RMS and is induced by MYOD through a looping enhancer

In a first step, we investigated *SKP2* levels across different tumor types finding that *SKP2* was strikingly overexpressed in RMS compared to control healthy skeletal muscle tissue and exhibited the highest levels of expression among all the adult and pediatric tumors analyzed (Fig. 1a). We found that *SKP2* mRNA levels were up-regulated in both FN-RMS and FP-RMS primary samples compared to skeletal muscle in two different RMS patients' cohorts[2,19], with the highest expression in the fusion-positive subtype (Fig. 1b and Supplementary Fig. 1a).

Moreover, the analysis of St. Jude Children's Research Hospital's PeCan Data Portal showed that *SKP2* expression in RMS patients had the highest levels among about 2500 pediatric cancers (Supplementary Fig. 1b). Analyzing a panel of cancer cell lines, RMS cells have the highest levels of *SKP2*, mirroring the results in RMS patients (Supplementary Fig. 1c). We confirmed SKP2 up-regulation at the protein level in 6 out of 7 FN-RMS and 5 out of 6 FP-RMS patients' samples *vs* control muscle tissues by immunohistochemistry (Fig. 1c). Similarly, SKP2 transcript and protein levels were markedly higher in a panel of patient-derived FN-RMS and FP-RMS cell lines compared to healthy human skeletal muscle myoblasts (HSMM), with the FP-RMS subtype showing the highest levels (Fig. 1d, e).

The potential biological importance of SKP2 in RMS was also supported by the effects of CRISPR/Cas9 knockout (KO) on the survival of RMS cell lines (Achilles project, https://depmap.org/portal/achilles). Both FP-RMS and FN-RMS subtypes showed SKP2 dependency, the latter being more dependent (<0.5 CERES score), suggesting a pro-survival role of SKP2, especially in the FN-RMS (Supplementary Fig. 1d).

Given the robust expression of SKP2 in RMS, we defined the chromatin status of regulatory regions of *SKP2* gene analyzing our previous ChIP-seq data for the active enhancers marker H3K27ac on a panel of 10 FP-RMS (6 cell lines and 4 primary tumors) and 9 FN-RMS (5 cell lines and 4 primary tumors)[5]. We found that an H3K27ac highly enriched region within the *SKP2* locus characterized both RMS subtypes compared to normal skeletal muscle (Fig. 1f).

The high levels of *SKP2* in RMS tumors and cell lines compared to the normal counterparts and the magnitude of H3K27ac occupancy at the *SKP2* locus suggested that its expression is highly transcriptionally regulated and presumably related to enhancers' deregulation. In addition, based on the observation that RMS has the highest *SKP2* levels among the analyzed tumors, we hypothesized that its transcription could be regulated by a lineage-specific transcription factor in RMS, such as MYOD, which is pro-tumorigenic in this subtype[7]. Thus, we investigated MYOD recruitment on the *SKP2* locus analyzing our previous ChIP-seq data in RD (FN-RMS) and RH4 (FP-RMS) cell lines[5,6]. We found that the H3K27ac-bound *cis*-regulatory intronic region was co-enriched with MYOD in both cell lines, supporting a potential role of MYOD in *SKP2* transcription (Fig. 2a). We also found a co-enrichment of PAX3-FOXO1 on the same region in RH4 cells, which had not been identified before (Fig. 2a). This suggests the presence of a core regulatory (CR) Transcription Factors (TF)s complex on the *SKP2* locus in FP-RMS containing both PAX3-FOXO1 and MYOD[20] and could propel the more elevated expression of SKP2 in FP-RMS compared to FN-RMS, and was in line with data showing that PAX3-FOXO1 regulates *SKP2* expression in FP-RMS[21,22].

In further support of a relationship between MYOD and SKP2, by analyzing publicly available data[19] we detected a significant positive correlation between *MYOD1* and *SKP2* expression both in FN-RMS and FP-RMS patients (Fig. 2b).

In addition, FN-RMS and FP-RMS cells express higher levels of MYOD compared to HSMM, except for the FN-RMS cell line RH2, with the highest levels in FP-RMS subtype (Supplementary Fig. 2a). MYOD depletion in two FN-RMS (RD and JR1) and two FP-RMS (RH4 and RH30) cell lines[23,24] using a previously validated siRNA against *MYOD1* (siMYOD)[5] resulted in a consistent decrease of SKP2 protein and mRNA levels (Fig. 2c–f). These results were confirmed using two additional independent MYOD siRNA and a CRISPR/Cas9 strategy to knock-out MYOD in RD and RH4 cells (Supplementary Fig. 2b–e).

To evaluate whether the H3K27ac bound intronic region at the *SKP2* locus could act as a looping enhancer, we examined its potential interaction with the *SKP2* promoter. Firstly, we performed HiChIP on RD FN-RMS cells and RH4 FP-RMS cells. We identified an interaction between the MYOD-bound intronic region and the *SKP2* promoter in both cell lines (Supplementary Fig. 3a). Interestingly, chromatin contacts appeared more enriched at *SKP2* promoter/MYOD-bound enhancer region in RD than in RH4 cells (Supplementary Fig. 3a, left). Conversely, RH4 cells showed more long-range chromatin contacts along *SKP2* locus (Supplementary Fig. 3a, right). Then, we performed chromosome conformation capture analysis (3 C) (Fig. 2g) on RD cells, examining in parallel RH4 cells as control, clearly validating the interaction between the two regions, indicative of a chromatin loop (Fig. 2h). Finally, HEK293T cells were co-transfected with a luciferase reporter plasmid carrying or not the identified MYOD-bound SKP2 intronic enhancer region (PGL3-SKP2 enh and PGL3-EV, respectively) and either a plasmid encoding the human *MYOD1* gene (FLAG-hMYOD1) or an empty vector (EV) as control (Supplementary Fig. 3b). SKP2 enhancer-driven luciferase activity was markedly up-regulated in FLAG-hMYOD1 cells 24 h post-transfection compared to control cells, confirming the ability of MYOD in transcriptionally activating the enhancer.

Analyzing public ChIP-seq data from human myoblasts and myotubes we found that MYOD bound the same intronic region identified in RMS, suggesting that MYOD could regulate *SKP2* in normal muscle cells as well (Supplementary Fig. 4a). Notably, MYOD binding was retained also in differentiated myotubes but the H3K27ac enrichment markedly decreased together with DNA accessibility, as demonstrated by reduced DNase-seq signal, indicating a more closed chromatin conformation potentially related to a lowering of transcriptional activity (Supplementary Fig. 4a). This is in line with the evidence that the MYOD binding alone is not sufficient for gene

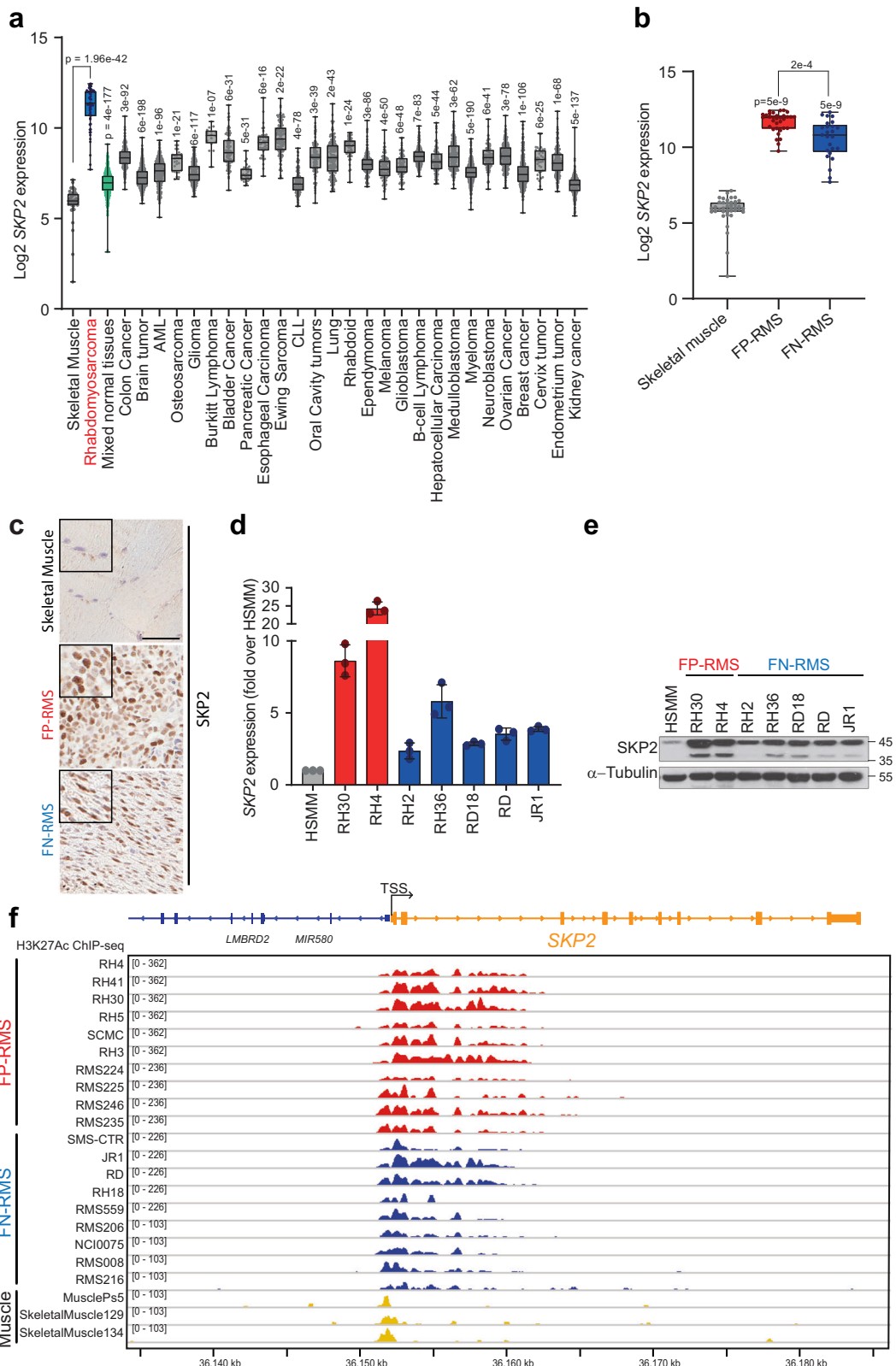

transactivation[25]. Consistently, the analysis of publicly available RNA-seq data of human myoblasts induced to differentiate in vitro showed that the transcript levels of *MYOD1* and *SKP2* increase within the first 24 h and then concomitantly decrease, returning to their starting levels and even lower in later phases of differentiation (Supplementary Fig. 4b). This, in contrast to *MYOG*, the main target gene of MYOD and a crucial inducer of myogenic differentiation, whose expression levels

remain elevated also by day 3 of differentiation (Supplementary Fig. 4b). Moreover, as shown in Supplementary Fig. 4c–e, retrovirus-mediated expression of exogenous MyoD in murine C3H/10T1/2 fibroblasts, a well described model of myogenic-like differentiation[26], resulted in increased mRNA and protein expression of SKP2 24 h post-infection in growth/proliferation medium (GM, supplemented with 10% serum) that was maintained 24 h upon shifting to differentiation

**Fig. 1 | SKP2 is highly expressed in RMS. a** Box plot depicting *SKP2* expression across skeletal muscle, rhabdomyosarcoma, mixed normal tissues and other human tumors (one-way ANOVA corrected with two-stage step-up method of Benjamini, Krieger and Yekutieli for multiple comparisons). AML Acute Myeloid Leukemia, CLL Chronic Lymphocytic Leukemia. Number of patients from each dataset is reported in Source Data File. Box plots show 25th to 75th quartiles, black bar shows the median, and whiskers go down to the smallest value and up to the largest. **b** Box plot of *SKP2* gene expression in a cohort of Fusion-Positive Rhabdomyosarcoma (FP-RMS) (*n* = 33) and Fusion-Negative Rhabdomyosarcoma (FN-RMS) (*n* = 25) patients (GSE66533) compared to skeletal muscle tissues (*n* = 40, GSE9103) (one-way ANOVA). Box plots show 25th to 75th quartiles, black bar shows the median, and whiskers go down to the smallest value and up to the largest.

**c** Representative SKP2 immunohistochemical staining in RMS primary tumors (FP-RMS *n* = 6, FN-RMS *n* = 7) and normal skeletal muscle (*n* = 2). Scale Bar = 100 μm. **d** mRNA levels (RT-qPCR) of *SKP2* in FP-RMS (red) and FN-RMS (blue) cell lines and in Human Skeletal Muscle Myoblasts (HSMM, gray; normal control cells) were normalized to *GAPDH* levels and reported as fold increase over HSMM values. *n* = 3 independent experiments, data presented as mean values ± SD. **e** Representative western blot (*n* = 3 independent experiments) of the indicated proteins in HSMM (normal control cells), FP-RMS and FN-RMS cell lines. α-TUBULIN is the loading control. **f** Representative profile of ChIP-seq read densities of H3K27ac at *SKP2* locus on a panel of FP-RMS (red), FN-RMS (blue) primary samples and cell lines and on skeletal muscle tissues (yellow). TSS Transcription start site. Source data are provided as a Source Data file.

medium (DM, serum-free medium). SKP2 expression revert to lower steady state levels by 48 h in DM, however, when the cells were fused into multi-nucleated structures that resembled muscle fibers. Therefore, while MYOD still bound to *SKP2* during terminal muscle differentiation, it is no longer able to induce *SKP2* expression.

Altogether, these findings indicate that *SKP2* is highly expressed in RMS and that MYOD concurs to its expression through direct binding of intronic enhancer elements.

### SKP2 depletion induces p27$^{Kip1}$-dependent cell cycle arrest reducing growth

Given that FN-RMS cells show one of the highest survival dependency for SKP2 in a CRISPR assay (Supplementary Fig. 1d), and its role in FN-RMS remains unknown, we defined the impact of SKP2 functions in this RMS subtype. To this end, we depleted SKP2 expression for 48 h in the FN-RMS cell lines RD and JR1 (both with mutated p53[24]), derived from recurrent and metastatic tumor samples, respectively. We noticed marked protein level up-regulation of the CDKIs p21$^{Cip1}$ and p27$^{Kip1}$ in SKP2 knocked-down cells compared to cells transfected with scrambled (control) siRNAs (siSCR) using either a validated individual siRNA sequence (Fig. 3a) or a pool of four different siRNAs (Supplementary Fig. 5a). Similar effects were observed in two additional FN-RMS cell lines, RD18 (p53 mutated) and RH36 (p53 wild-type), both derived from metastatic samples[24] (Supplementary Fig. 5b). The increase of *CDKN1A* and *CDKN1B* transcripts, encoding p21$^{Cip1}$ and p27$^{Kip1}$, respectively, did not uniformly correlate with protein levels in all cell lines at this time point (Fig. 3b and Supplementary Fig. 5c). However, at earlier time points (<24 h) p21$^{Cip1}$ and p27$^{Kip1}$ protein levels were up-regulated prior to mRNA induction in all four cell lines (Supplementary Fig. 5d, e) demonstrating that SKP2 affects the CDKIs levels at least in part in a post-translational manner. Accordingly, both p21$^{Cip1}$ and p27$^{Kip1}$ basal protein levels increased in RD and JR1 cells 8 h post-treatment with the proteasome inhibitor MG132 (Supplementary Fig. 5f), indicating that they are modulated in a proteasome-dependent manner in FN-RMS, as reported in myoblasts[27].

We then determined whether SKP2 directly targets p21$^{Cip1}$ and p27$^{Kip1}$ for proteasomal degradation in FN-RMS cells. Co-immunoprecipitation (Co-IP) of endogenous SKP2 showed that SKP2 directly interacts with p27$^{Kip1}$ in RD and JR1 cells (Fig. 3c), which is confirmed by Proximity Ligation assay (PLA) (Fig. 3d). Conversely, no interaction was detected between SKP2 and p21$^{Cip1}$ in either a direct or a reciprocal Co-IP and a PLA assay (Supplementary Fig. 5g, h).

Consistent with a key role of SKP2 on cell cycle progression, a 48 h knockdown of SKP2 enhanced the percentage of FN-RMS cells in G1 phase (27 ± 6%, 14 ± 4%, 23 ± 3% and 11 ± 0.6% increase) while lowered that in S phase (37 ± 6%, 33 ± 5%, 31 ± 1% and 31 ± 4% decrease) for RD, JR1, RD18 and RH36, respectively, *vs* scrambled siRNA cells (Fig. 3e, f and Supplementary Figs. 5i, j).

To verify the role of p27$^{Kip1}$ up-regulation caused by SKP2 silencing on cell proliferation, we performed rescue experiments. *CDKN1B*/p27$^{Kip1}$ silencing alone did not affect basal proliferation rate of RD and JR1 cells but it counteracted the anti-proliferative effect of SKP2

depletion maintaining cell growth of SKP2 knocked-down cells to values similar to those in scrambled siRNA control cells (Fig. 3g, h). Depletion of p27$^{Kip1}$ also reduced p21$^{Cip1}$ increase in SKP2-silenced cells with an indirect mechanism that needs to be clarified.

We, then, sought to evaluate whether pharmacological inhibition of SKP2 could have similar effects. To this end, we treated FN-RMS cells with the compound SMIP004, a SKP2 inhibitor that promotes SKP2 degradation and stabilizes p27$^{Kip1}$ reducing tumor cells survival in vitro[28]. Treatment with SMIP004 mirrored SKP2 genetic depletion increasing p21$^{Cip1}$ and p27$^{Kip1}$ protein and mRNA levels and lowering proliferation by enhancing G1 phase (26 ± 5% and 31 ± 4% increase) and decreasing S phase (31 ± 5% and 35 ± 6% decrease) cells percentage in RD and JR1 cells, respectively (Fig. 3i–m). To verify that the response to SKP2 inhibition represented a tumor vulnerability of FN-RMS cells, a panel of normal cells was treated with the agent in a dose response assay, as well. As shown in Supplementary Fig. 5k, human healthy myoblasts (considered the normal counterpart of RMS cells) and lung and dermal fibroblasts were considerably less sensitive to SMIP004 compared to FN-RMS cells.

Collectively, these results suggest that SKP2 promotes FN-RMS cell proliferation at least in part by directly targeting p27$^{Kip1}$ for degradation and validate SKP2 as a survival vulnerability in FN-RMS.

### SKP2 suppression leads to myogenic differentiation by increasing p57$^{Kip2}$

The defective ability of FN-RMS cells to differentiate highly contributes to tumorigenesis[29]. Notably, cell cycle arrest is a prerequisite for myogenic differentiation[30]. Moreover, SKP2 promotes the maintenance of an undifferentiated state in normal[31–33] and in colon cancer cells[34]. Thus, we evaluated the role of SKP2 in the differentiation of FN-RMS cells. Forty-eight hours of SKP2 siRNA-mediated knockdown increased both protein and transcript levels of one of the master myogenic TFs, MYOG, compared to scrambled siRNA in all the four examined cell lines (Fig. 4a, b and Supplementary Fig. 6a, b). The protein levels of MYOD increased alongside those of the CDKI p57$^{Kip2}$, which promotes myogenesis[35] and was reported to be a SKP2 direct target[36] (Fig. 4a, and Supplementary Fig. 6a). Notably, the correlation between transcript and protein levels did not appear to be uniform in all cell lines highlighting, once more, both direct and indirect effects of *SKP2* knockdown (Fig. 4b and Supplementary Fig. 6b). Moreover, although FN-RMS cells were cultured in growth/proliferation medium (GM, supplemented with 10% serum), SKP2 depletion was able to induce de novo expression of the terminal muscle differentiation marker Myosin Heavy Chain (*MYH2*/MyHC) 4 days after transfection, which was followed by the appearance of elongated multinucleated MyHC-positive myofiber-like structures 2 days later (Fig. 4c, d and Supplementary Fig. 6c, d). A comparison of SKP2 siRNA treated cells to their scrambled siRNA treated counterparts indicated that the percentage of MyHC positive cells increased by approximately 7-fold, 11-fold, 21-fold and 13-fold in RD, JR1, RD18, and RH36, respectively (Fig. 4e and Supplementary Fig. 6e).

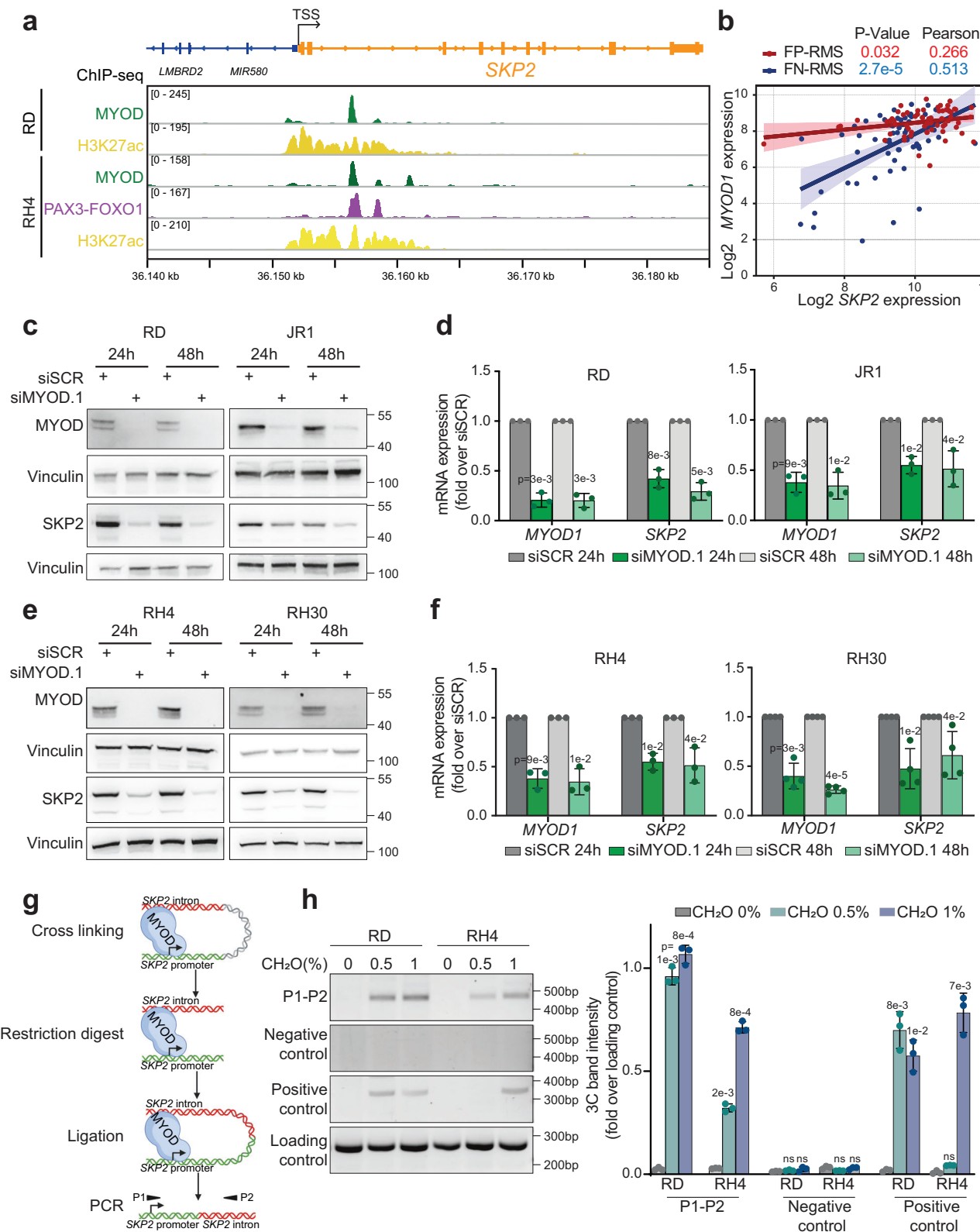

Similar outcomes were observed in RD and JR1 cells by employing an orthogonal method for silencing *SKP2* using two lentiviral vectors expressing individual short hairpin (sh)RNAs against *SKP2* (shSKP2.1 and shSKP2.2). Here as well, SKP2 shutdown promoted increased expression of p21$^{Cip1}$, p27$^{Kip1}$, MYOD, MYOG and p57$^{Kip2}$ compared to non-targeting control shRNA (shSCR) (Fig. 4f, g) and expanded the ratio of MyHC positive cells in both, RD and JR1 cells by at least 12-fold

for both shRNAs (Fig. 4h). The observed extent of MyHC induction was consistent with our previous results obtained in another differentiation model cultured in growth/proliferation medium[6]. Inhibition of SKP2 using SMIP004 also up-regulated MYOD, MYOG and p57$^{Kip2}$ (Fig. 4i, j) and markedly increased the expression of *MYH2*/MyHC and the number of MyHC positive RD and JR1 cells compared to vehicle (Fig. 4k–m).

**Fig. 2 | SKP2 is regulated by a MYOD bound enhancer. a** Representative profile of ChIP-seq read densities of MYOD (green), H3K27ac (yellow) and PAX3-FOXO1 (red) at *SKP2* locus on RD (FN-RMS) and RH4 (FP-RMS) cells. TSS, Transcription start site. **b** Pearson's correlation analysis of *MYOD1* versus *SKP2* expression in FN-RMS (*n* = 58) and FP-RMS (*n* = 54) primary tumors. Line represents linear regression of data. Pearson's correlation and *p* values are reported in the figure. *P*-value was calculated by a two-tailed Pearson correlation test. Error bands represent the 95% confidence interval for the Loess curve fit of the data. **c, e** Representative western blot (*n* = 3 independent experiments) of the indicated proteins on RD and JR1 FN-RMS cells transfected with either Scrambled (siSCR) or MYOD siRNA (siMYOD.1) at 24 h (h) and 48 h post-transfection. Vinculin is the loading control. **d, f** mRNA levels (RT-qPCR) of *MYOD1* and *SKP2* on cells treated as in (**c, e**) were normalized to *GAPDH* levels and expressed as fold increase over siSCR. *n* = 3 (RD, JR1, RH4) and *n* = 4 (RH30) independent experiments, data presented as mean values ± SD, Student's two-tailed *t*-test. **g** Workflow schematic model of Chromosome Conformation Capture (3 C) assay. Created with BioRender.com. **h** (left) Representative electrophoretic run and (right) band intensity quantification of amplification products from 3 C assay on RD (FN-RMS) and RH4 (FP-RMS) cells. P1-P2, primers were designed on *SKP2* promoter and intronic enhancer, respectively (see **g**). As loading control a region lacking restriction sites at *SKP2* enhancer has been used. As 3 C positive control we used MYOD looping between *SNAI2* distal enhancer and promoter as described in our previous work. A negative control was selected as reported in Supplementary Methods. *n* = 3 independent experiments, data presented as mean values ± SD, one-way ANOVA. Source data are provided as a Source Data file.

We then found that p57[Kip2] was modulated by proteasomal degradation in RD and JR1 cells treated with the proteasome inhibitor MG132 for 8 h (Fig. 5a). To verify whether SKP2 could directly concur to p57[Kip2] proteasomal degradation we evaluated the interaction between the two proteins. Both Co-immunoprecipitation of endogenous SKP2 and PLA assay highlighted a SKP2-p57[Kip2] direct interaction in both RD and JR1 cells (Fig. 5b, c). To investigate the role of SKP2-dependent p57[Kip2] regulation in the differentiation of FN-RMS cells, we co-depleted *CDKN1C*/p57[Kip2] and SKP2 by siRNA as rescue experiment (Fig. 5d). Simultaneous down-regulation of both molecules in both RD and JR1 cells completely prevented the formation of MyHC-positive multinucleated structures seen after SKP2 silencing alone (Fig. 5e, f). Molecularly, SKP2 silencing, as in Fig. 4a, up-regulated both p57[Kip2] and MYOD levels. Moreover, it resulted in a reduction of the ratio between the levels of MYOD phosphorylated on Serine-200 (MYOD[S200]), which is degraded by the proteasome[37], and MYOD total form *vs* scrambled siRNA (Fig. 5d). This suggests increased MYOD stability. The depletion of p57[Kip2] in SKP2-silenced cells enhanced the MYOD[S200]/MYOD ratio with respect SKP2 siRNA alone (Fig. 5d). The result was the inhibition of MYOD increase upon SKP2 suppression, and it strongly suggests that p57[Kip2] is needed to promote this phenomenon by preventing MYOD degradation.

Notably, even under basal conditions p57[Kip2] depletion alone was sufficient to increase MYOD[S200]/MYOD ratio *vs* scrambled siRNA indicating that the CDKI is, at least in part, active in this condition (Fig. 5d).

Altogether, these findings indicate that, upon SKP2 depletion in FN-RMS cells, p57[Kip2] protects MYOD from Serine-200 phosphorylation and subsequent proteasomal degradation, thus concurring to differentiation, consistent with data in differentiating muscle cells[35,37]. Forced lentiviral-mediated p57[Kip2] expression (pLENTIp57) in RD and JR1 cells led to MYOD and MYOG transcript and protein levels up-regulation compared to empty control vector (EV) (Supplementary Fig. 7a, b). However, despite the induction of *MYH2*/MyHC expression and enhancement of MyHC positive p57[Kip2]-overexpressing tumor cells, the extent of differentiation was modest with respect to that shown by SKP2-depleted cells (Supplementary Fig. 7c–e), suggesting that SKP2 silencing provides a pro-differentiation cell context in which p57[Kip2] highly support the myogenic program.

Next, we validated the anti-differentiation function of SKP2 in physiological conditions using untransformed C2C12 murine myoblasts, a well-known in vitro myogenesis model. Murine Skp2 decreased in cells shifted to differentiation medium (DM), consistent with previous data[27] (Supplementary Fig. 8a). Further, Skp2 down-regulation accelerated myoblasts fusion that was evident as early as 2 days after the shift to DM compared to control scrambled siRNA (Supplementary Fig. 8b, c). Conversely, retroviral-mediated expression of GFP-tagged SKP2 in human myoblasts prevented MYOG up-regulation induced by 48 h of culture in DM compared with a GFP control vector, and blunted the increase of p27[Kip1] and p57[Kip2] as well as the transcriptional induction of *MYH2* (Supplementary Fig. 8d, e). Consistently, human SKP2-expressing myoblasts were largely unable

to differentiate and fuse into multinucleated MyHC-positive fibers (Supplementary Fig. 8f).

To evaluate the indirect effects of *SKP2* silencing on gene expression we performed RNA-seq followed by gene set enrichment analysis (GSEA) on RD and JR1 cells in which SKP2 was depleted by shSKP2.2 *vs* control shSCR cells. The differentially expressed genes were reported in Supplementary Data 1.

A significant positive enrichment in myogenesis and cell differentiation, muscle contraction and targets of MYOD pathways was noticed in both shSKP2.2 cell lines (Supplementary Fig. 9a). GSEA analysis also showed significant negative enrichment for genes down-regulated in senescence, for stemness genes as well as for cell cycle genes included in several gene sets (Supplementary Fig. 9a). Among the top 20 myogenic up-regulated genes (Supplementary Fig. 9a, b and Supplementary Data 2) were included those that have been reported as induced in vitro during the transition from proliferation to differentiation in myoblasts[38,39]. The top 10 genes down-regulated in senescence and the top 20 stemness genes expressed in murine stem cells are shown in Supplementary Fig. 9c, d and Supplementary Data 2.

Then, we evaluated the involvement of MYOD in the transcriptional pro-myogenic landscape induced by SKP2 knockdown performing chromatin immunoprecipitation followed by DNA sequencing (ChIP-seq) for MYOD and the enhancer mark H3K27ac on RD shSKP2.2 and control shSCR cells. Analysis of MYOD peaks showed that global MYOD genome-wide binding was slightly enhanced after SKP2 depletion (Supplementary Fig. 9e). This suggests focused enhanced binding at MYOD peaks rather than MYOD de novo binding, as already shown by others and our group[6,39]. Moreover, we found a marked increase of H3K27ac binding at MYOD-bound sites (Supplementary Fig. 9e). We then identified 16 myogenic genes up-regulated in both RD and JR1 SKP2-depleted cells that are MYOD targets being common to MYOD-TARGETS_UP and MYOGENESIS gene sets (Supplementary Fig. 9f). Analyzing MYOD peaks at *CDKN1A*, *MYL1*, *MYOG,* and *MYBPH* genes regions we found that MYOD deposition slightly increased in parallel with a marked enhancement of H3K27ac deposition and was associated to a remarkable induction of gene expression (Supplementary Fig. 9g). Of note, the enhancer in *MYBPH* gene also acts as enhancer on *MYOG*, as reported[6].

Altogether, these data indicate that SKP2 prevents a differentiation program in FN-RMS cells mainly by down-regulating the protein levels of p57[Kip2] leading to MYOD degradation, and by indirectly hindering the transactivation ability of MYOD on myogenic genes.

## SKP2 silencing reduces stemness and tumorigenicity in vitro and suppresses growth in vivo

Then, we evaluated the functional effects of SKP2 depletion on the tumorigenic properties of RD and JR1 cells. *SKP2* shRNA silencing with shSKP2.1 and shSKP2.2 in FN-RMS cells strongly inhibited tumor-specific ability to form anchorage-independent colonies in soft agar 20 days after seeding compared to scrambled shRNA (shSCR) (Fig. 6a–c). Since SKP2 has been associated with stem-like

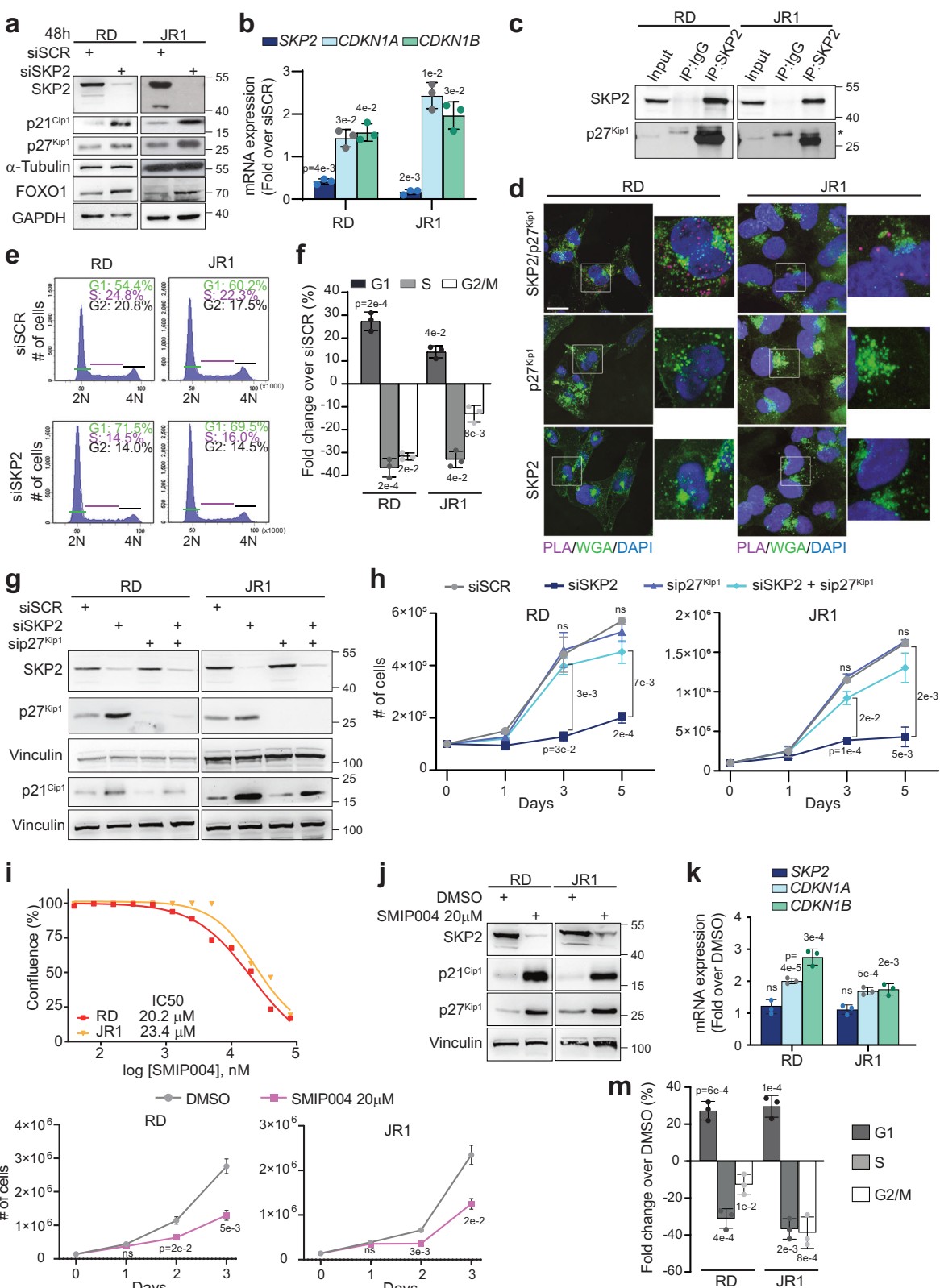

characteristics of cancer cells[40] and our RNA-seq data indicate a role of SKP2 in stemness in FN-RMS, we assessed the stem-related features of FN-RMS cells measuring their sphere-forming ability when cultured in stemness medium[6] (see Methods). *SKP2* knockdown greatly lowered the number of rhabdospheres 20 days post-seeding compared to shSCR (Fig. 6d, e). The same inhibitory effects on anchorage-

independent growth and sphere-forming ability were observed after treatment with the SKP2 inhibitor SMIP004 (Supplementary Fig. 10a–d).

Furthermore, in agreement with the transcriptional effects detected, siRNA-mediated SKP2 depletion for 48 h induced senescence-associated β-galactosidase activity increasing the

**Fig. 3 | SKP2 depletion induces p27$^{Kip1}$-dependent cell cycle arrest reducing growth. a** Representative western blot on cells transfected with Scrambled (siSCR) or SKP2 siRNA (siSKP2). α-TUBULIN is the loading control. **b** mRNA levels of *SKP2, CDKN1A* and *CDKN1B* on cells treated as in (**a**) were reported as fold increase over siSCR (1 arbitrary unit, not reported). Data presented as mean values ± SD, Student's two-tailed *t*-test. **c** Representative western blot of co-Immunoprecipitation of endogenous SKP2. (*) indicates IgG. **d** Representative confocal pictures of Proximal Ligation Assay (PLA) for SKP2 and p27$^{Kip1}$. Proximity/interactions are visualized as red dots (upper). DAPI (blue) stained nuclei and WGA (green) membrane glycoproteins. p27$^{Kip1}$ (middle) or SKP2 (lower) antibodies alone were PLA negative controls. Scale Bar = 10 μm. **e** Representative flow cytometry cell cycle diagrams of cells treated as in (**a**). **f** Histogram depicts fold changes of the percentage of siSKP2-cells in cell cycle phases over siSCR-cells (1 arbitrary unit, not reported). Data presented as mean values ± SD, Student's two-tailed *t*-test. **g** Representative western blot on cells transfected with SCR, SKP2, p27$^{Kip1}$ or SKP2 +

p27$^{Kip1}$ siRNA 72 h post-transfection. Vinculin is the loading control. **h** Growth curve analysis of cells treated as in (**g**). Data presented as mean values ± SD, two-way ANOVA (upper siSKP2+sip27$^{Kip1}$ *vs* siSCR, middle siSKP2+sip27$^{Kip1}$ vs siSKP2, lower siSKP2 vs siSCR). **i** Dose-response curves of cells treated with increasing concentration of SMIP004. Data presented as mean values ± SEM. **j** Representative western blot on cells treated with SMIP004 20 μM for 48 h. Vinculin is the loading control. **k** mRNA levels of *SKP2, CDKN1A* and *CDKN1B* on cells treated as in (**j**) were reported as fold increase over DMSO (1 arbitrary unit, not reported). Data presented as mean values ± SD, Student's two-tailed *t*-test. **l** Growth curve analysis of cells treated as in (**j**). Data presented as mean values ± SD, Student's two-tailed *t*-test. **m** Histogram depicts fold changes of the percentage of SMIP004-treated cells in cell cycle phases over DMSO cells (1 arbitrary unit, not reported). Data presented as mean values ± SD, Student's two-tailed *t*-test. All the presented data derive from *n* = 3 independent experiments. Source data are provided as a Source Data file.

percentage of β-galactosidase-positive senescent cells *vs* scrambled siRNA in RD and JR1 cells (Fig. 6f, g) suggesting an involvement of SKP2 in suppressing senescence in FN-RMS.

Next, we evaluated whether SKP2 promoted tumor growth by subcutaneously engrafting JR1 cells expressing shSKP2.1, shSKP2.2, or scrambled shRNA (shSCR) as control. SKP2 knockdown strongly delayed the appearance of tumor masses and strikingly reduced tumor size and weight compared with tumors from scrambled shRNA cells in the same mice (Fig. 6h–j). Moreover, SKP2-depleted explanted tumors displayed increased positivity for p27$^{Kip1}$, p57$^{Kip2}$ and MYOG and decreased positivity for the proliferative marker Ki67 associated to de novo appearance of MyHC-positive cells *vs* scrambled shRNA tumors (Fig. 6k).

Similarly, shSKP2.2-silenced RD cells gave rise to tumors significantly smaller compared to scrambled shRNA tumors, which showed increased staining for p27$^{Kip1}$, p57$^{Kip2}$ and MYOG and reduced Ki67 expression (Supplementary Figs. 10e–h).

Altogether, these results support a role for SKP2 in tumorigenic properties of FN-RMS cells in vitro and in vivo.

## NEDDylation inhibition prevents SKP2 functions and suppresses tumor growth

Going forward, we wanted to investigate the possible therapeutic implications of this specific SKP2 dependency in FN-RMS. As no specific direct SKP2 inhibitors have yet been developed in clinic, we investigated the effectiveness of indirect SKP2 inhibition. The SKP2-containing SCF/CRL1 complex strictly relies on NEDDylation for activity[41]. Specifically, NEDD8-activating enzyme (NAE) is responsible for the ubiquitin-like peptide NEDD8 binding to Cullin1 to activate the complex promoting SKP2 action[41]. Therefore, we used the NAE inhibitor MLN4924 (Pevonedistat) to indirectly block SKP2 functions[17]. RD and JR1 cells treated with MLN4924 in a dose-response assay (Supplementary Fig. 11a) exhibited reduction of the NEDDylated Cullin1 and strong induction of p21$^{Cip1}$ and p27$^{Kip1}$ protein levels (Fig. 7a). Functionally, MLN4924 used at the GI$_{50}$ dose significantly decreased the proliferation of FN-RMS cells (Fig. 7b). As observed for the SKP2 inhibitor SMIP004, normal healthy myoblasts and fibroblasts from lung and dermis displayed lower sensitivity to the drug respect to RD and JR1 cells suggesting a FN-RMS vulnerability to SKP2 inhibition (Supplementary Fig. 11b).

Despite up-regulation of p57$^{Kip2}$ and MYOD protein levels, as observed after SKP2 silencing, MLN4924 down-regulated MYOG expression, conversely to what happens after SKP2 knockdown (Fig. 7c). This result is in agreement with data showing that MLN4924 suppresses murine myoblasts differentiation partly preventing MYOG expression[42]. Treatment with MLN4924 enhanced the percentage of β-galactosidase-positive senescent cells compared to vehicle (DMSO) as soon as 48 h post-treatment (Fig. 7d, e) and affected the in vitro tumorigenic abilities of RD and JR1 cells to grow as colonies

in soft agar and to form stem-like rhabdospheres (Fig. S11c–f). We then explored whether decreased growth of treated cells was due to the cells undergoing cell death instead of differentiation and noticed a significant increase of Caspase 3/7 activation (Fig. 7f). We also used MLN4924 to treat patient-derived orthotopically xenografted (PDX) FN-RMS cells, one of which mutated for p53 (B011_YC) and another expressing p53 wild-type (from two different tumor sites: B012_YC and B012_ZC)[43]. MLN4924 treatment resulted in marked decrease in cell viability/survival and induction of p21$^{Cip1}$ and p27$^{Kip1}$ in o-PDX cells (Supplementary Fig. 11g, h).

Next, we investigated the effects of NEDDylation inhibition on tumor growth. RD cells were subcutaneously inoculated into immune-deficient mice and, once tumors were palpable (about 200 mm$^3$ at Day 1), animals were injected with MLN4924 (50 mg/kg, subcutaneously) once a day for 6 days a week for 3 weeks, and tumor growth was assessed, a protocol similar to those used in other cancers[17]. MLN4924 completely prevented the growth of tumor masses, and in 2 out of 5 mice even induced a volumetric reduction of the initial mass, resulting in 1.0 ± 0.3 *vs* 3.5 ± 0.8 fold-increase in MLN4924 *vs* control vehicle (DMSO)-treated tumors at the end of the experiment over Day 1 (Fig. 7g, h). Drug-treated tumors also displayed striking weight reduction compared to controls (Fig. 7i). Tumor sections from MLN4924-treated mice showed increased immunohistochemical p21$^{Cip1}$ and p27$^{Kip1}$ staining associated to enhanced positivity for cleaved-Caspase3 and reduced Ki67 expression compared to vehicle (Fig. 7j).

MLN4924 treatment also significantly affected tumor growth of JR1 xenografts for which treatment was started when tumors reached approximately twice the size as RD xenografts, i.e., about 400 mm$^3$ at Day 1. The volume of tumor masses showed 2.9 ± 0.6 vs 5.7 ± 0.7 fold-increase for MLN4924 *vs* vehicle at the end of the experiment over Day 1 (Supplementary Fig. 11i, j) and was associated with lower tumor weight in the MLN4924- compared to vehicle-treated mice (Supplementary Fig. 11k). Even in this model, drug treatment induced up-regulation of p21$^{Cip1}$, p27$^{Kip1}$, activation of Caspase 3/7 and reduction of Ki67 staining (Supplementary Fig. 11l).

Collectively, these findings indicate that inhibition of NEDDylation compromises SKP2 functions up-regulating p27$^{Kip1}$ affecting proliferation and cell viability and hampering the in vivo tumorigenic properties of FN-RMS cells.

## Discussion
SKP2 is an oncogenic E3 ubiquitin ligase often overexpressed and/or amplified in cancer[12]. SKP2 acts post-translationally as a component of the SCF/CRL1 complex binding specific proteins promoting their ubiquitylation and subsequent degradation. In particular, its oncogenic effects have been linked to p27$^{Kip1}$ degradation resulting in cell cycle progression and escape from senescence[44,45]. Several studies indicate that blockade of SKP2 functions arrests tumorigenesis in vitro

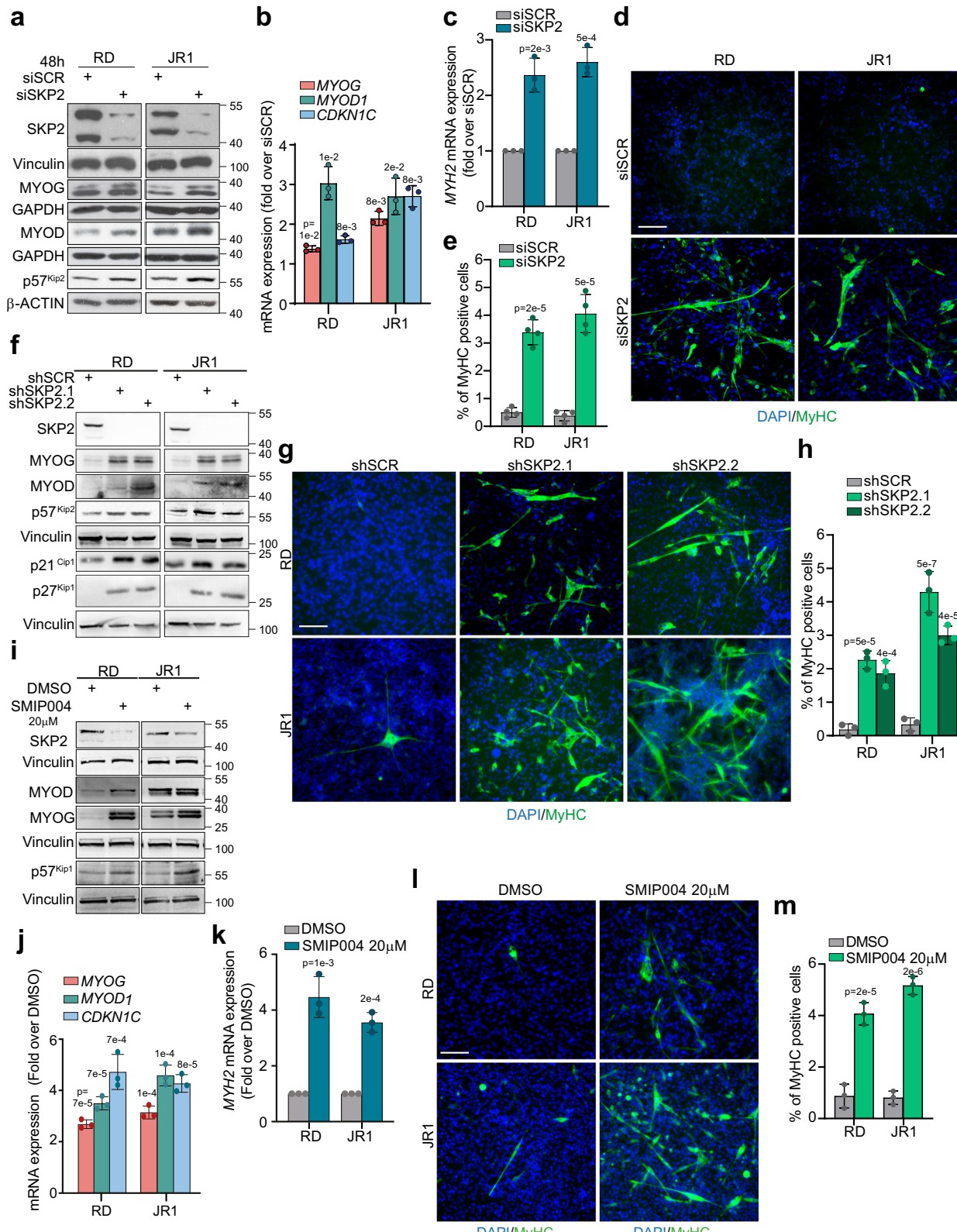

and in vivo suggesting that its targeting can have translational relevance[16–18,45].

Here, we unveil a pro-tumorigenic MYOD-SKP2 axis in FN-RMS through which MYOD, which induces skeletal muscle differentiation in physiological conditions, supports the sustained expression of SKP2 that, in turn, promotes growth and prevents differentiation.

This study demonstrates that FN-RMS is dependent on SKP2 for tumorigenicity, which provides a strong susceptibility to SKP2 inhibition. We report that RMS primary samples show higher *SKP2* levels than normal skeletal muscle tissue and the highest among all the interrogated adult and pediatric cancers. Previously, SKP2 was shown modulated by PAX3-FOXO1 in FP-RMS cells in which its suppression

**Fig. 4 | SKP2 suppression leads to myogenic differentiation. a** Representative western blot on cells transfected with either Scrambled (siSCR) or SKP2 siRNA (siSKP2). Vinculin, GAPDH and β-Actin are the loading controls. **b** mRNA levels of *MYOG*, *MYOD1* and *CDKN1C* on cells treated as in (**a**) were reported as fold increase over siSCR (1 arbitrary unit, not reported). Data presented as mean values ± SD, Student's two-tailed *t*-test. **c** mRNA levels of *MyH2* on cells treated as in (**a**) in growth medium (GM) for 4 days post-transfection were expressed as fold increase over siSCR (1 arbitrary unit). Data presented as mean values ± SD, Student's two-tailed *t*-test. **d** Representative immunofluorescence on cells treated as in (**a**) for 6 days in GM showing expression of Myosin Heavy Chain (MyHC) (green). DAPI (blue) stained nuclei. Scale Bar = 100 μm. **e** Histogram depicts the quantification of MyHC-positive cells treated as in (**d**). Data presented as mean values ± SD, Student's two-tailed *t*-test. **f** Representative western blot on cells infected with either shSCR, shSKP2.1 or shSKP2.2. Vinculin is the loading control. **g** Representative immunofluorescence on cells treated as in (**f**) for 6 days, showing expression of MyHC

(green). DAPI (blue) stained nuclei. Scale Bar = 100 μm. **h** Histogram depicts the quantification of MyHC-positive cells treated as in (**g**). Data presented as mean values ± SD, two-way ANOVA. **i** Representative western blot on cells treated with SMIP004 20 μM for 168 h. Vinculin is the loading control. **j** mRNA levels of *MYOG*, *MYOD1* and *CDKN1C* on cells treated as in (**i**) were reported as fold increase over DMSO (1 arbitrary unit, not reported). Data presented as mean values ± SD, Student's two-tailed *t*-test. **k** mRNA levels of *MyH2* on cells treated as in (**i**), cultured in GM, were expressed as fold increase over DMSO (1 arbitrary unit). Data presented as mean values ± SD, Student's two-tailed *t*-test. **l** Representative immunofluorescence of cells treated as in (**i**) showing expression of MyHC (green). DAPI (blue) stained nuclei. Scale Bar = 100 μm. **m** Histogram depicts the quantification of MyHC-positive cells treated as in (**i**). Data presented as mean values ± SD, Student's two-tailed *t*-test. All presented data derive from *n* = 3 independent experiments. Source data are provided as a Source Data file.

leads to proliferation inhibition associated to p27[Kip1] up-regulation[21,22]. However, FN-RMS cell lines are specifically sensitive to SKP2 knockout in a CRISPR/Cas9 survival screen compared to FP-RMS cells and other cancer cell lines. Consistently, we demonstrate that SKP2 depletion halts FN-RMS cells proliferation and promotes differentiation, impairing their tumorigenic features in vitro and in vivo.

Since no amplification of the *SKP2* 5p13 gene region has been found in RMS patients[3], we hypothesized that the mechanism responsible for the high expression of SKP2 in FN-RMS may be transcriptional, possibly involving a lineage-related factor selectively expressed in the tumor.

We identified an intronic regulatory region highly enriched for the enhancers' histone mark H3K27ac within the *SKP2* locus of RMS tumor samples and cell lines compared to normal muscle cells. We found that MYOD is co-enriched on this intronic enhancer region and can directly interact with the *SKP2* promoter in both FN- and FP-RMS cells. Moreover, we provide evidence of *SKP2* promoter transactivation in a MYOD-dependent manner by the identified enhancer region. We also identify a PAX3-FOXO1 binding region near the MYOD-bound *SKP2* enhancer in FP-RMS cells suggesting that both MYOD and PAX3-FOXO1 participate in the induction of *SKP2* in this tumor subtype, possibly as components of a core transcription TFs complex[20].

MYOD was validated as a regulator of SKP2 expression through loss-of-function and gain-of-function experiments, in line with the positive correlation found between *MYOD1* and *SKP2* RNA expression levels in RMS patient samples. Additionally, we report that the same *SKP2* enhancer region is bound by MYOD in myoblasts and differentiated myotubes, further corroborating the idea of a lineage-specific regulation of *SKP2*. Notably, although MYOD remains bound in differentiated myotubes, we found a decrease of both H3K27ac enrichment and DNA accessibility compared to myoblasts suggesting less accessible chromatin upon differentiation. This finding is in agreement with the modulation of both *MYOD1* and *SKP2* expression in myoblasts, which increases in parallel during early differentiation to drop to basal or even lower levels at a late phase presumably to allow the cells to exit the cell cycle. Altogether, these data support a model whereby MYOD contributes to *SKP2* expression in normal myoblasts and RMS cells through an intronic enhancer whose MYOD-mediated activity declines during normal muscle differentiation but persists in RMS, supporting sustained SKP2 expression (Fig. 8a). This is consistent with the pro-tumorigenic transcriptional functions of MYOD in FN-RMS reported by us and other groups[6,7].

We demonstrate that SKP2 sustains FN-RMS cell cycle progression by directly binding to and facilitating the degradation of p27[Kip2]. SKP2 genetic suppression results in p27[Kip2]-dependent G0/G1 cell cycle arrest. However, conversely to what reported in myxofibrosarcoma (MFS) and undifferentiated pleomorphic sarcoma (UPS)[17], in our cancer context the effects of SKP2 inhibition appear to be, at least in part, unrelated to the mutational status of p53 and pRb since similar

functional effects were seen in both p53-mutated and p53 wild-type FN-RMS cells and PDX.

Moreover, since p27[Kip1] can also be regulated by p53-dependent ubiquitin ligases[18] but its depletion alone is unable to enhance basal cell proliferation in p53-mutated FN-RMS cells with intact SKP2 expression, our data suggest that SKP2 is the primary factor maintaining p27[Kip1] protein levels under a functional threshold in our tumor model.

Our results also uncover an unprecedented role for SKP2 in regulating myogenic-like differentiation in FN-RMS. We show that SKP2 knockdown up-regulates MYOD, MYOG and p57[Kip2] levels allowing de novo expression of terminal differentiation marker MyHC and formation of multinucleated muscle-like structures. The anti-differentiation role of SKP2 is also consistent with our findings showing that Skp2-depleted normal murine myoblasts undergo anticipated differentiation while human myoblasts continuously expressing SKP2 are unable to fuse and differentiate. Our data support a model in which SKP2 prevents differentiation in FN-RMS cells by directly binding to and promoting the degradation of p57[Kip2]. When SKP2 is suppressed, the up-regulated p57[Kip2] stabilizes at least in part MYOD by reducing its phosphorylation at Serine-200 and its consequential degradation thus facilitating differentiation, consistent with the active role of p57[Kip2] during normal myogenesis[35,37]. The concomitant mRNA increase in MYOD levels we noticed after SKP2 knockdown in two cell lines may be mediated at least in part by transcriptional auto-induction[46]. However, forcing p57[Kip2] expression alone in the presence of SKP2 enhances MYOD protein and mRNA levels and facilitates differentiation but with a markedly reduced effect compared to decreasing SKP2 alone. In agreement, MYOD increase per se is insufficient to trigger cell differentiation in FN-RMS[47] but, conversely, its continuous expression is required for tumor cell survival[6,7]. Therefore, SKP2 depletion allows MYOD to acquire the capability to transactivate *MYOG* and a number of myogenic genes, in line with enhanced binding of MYOD and H3K27ac on their regulatory regions and the evidence of increased H3K27ac global deposition at the MYOD-bound regions suggesting transcriptional activation. Thus, our results identify SKP2 levels reduction as a critical step of myogenic differentiation in FN-RMS through the selective transcriptional induction of a number of genes partly MYOD targets included in the myogenesis signatures[48] and specifically expressed in committed muscle progenitors[49].

Furthermore, SKP2 concurs to the expression of a number of genes involved in stemness in agreement with marked reduction of rhabdospheres formation ability in stem cell medium after its genetic and pharmacologic inhibition.

SKP2 also maintains active the expression of genes that are down-regulated during senescence. The senescent phenotype observed in a subpopulation of SKP2-depleted FN-RMS cells is consistent with a role of p27[Kip1], which induces p53-independent senescence after Skp2 loss in murine prostate tumors[45], and of p57[Kip2], which promotes cell

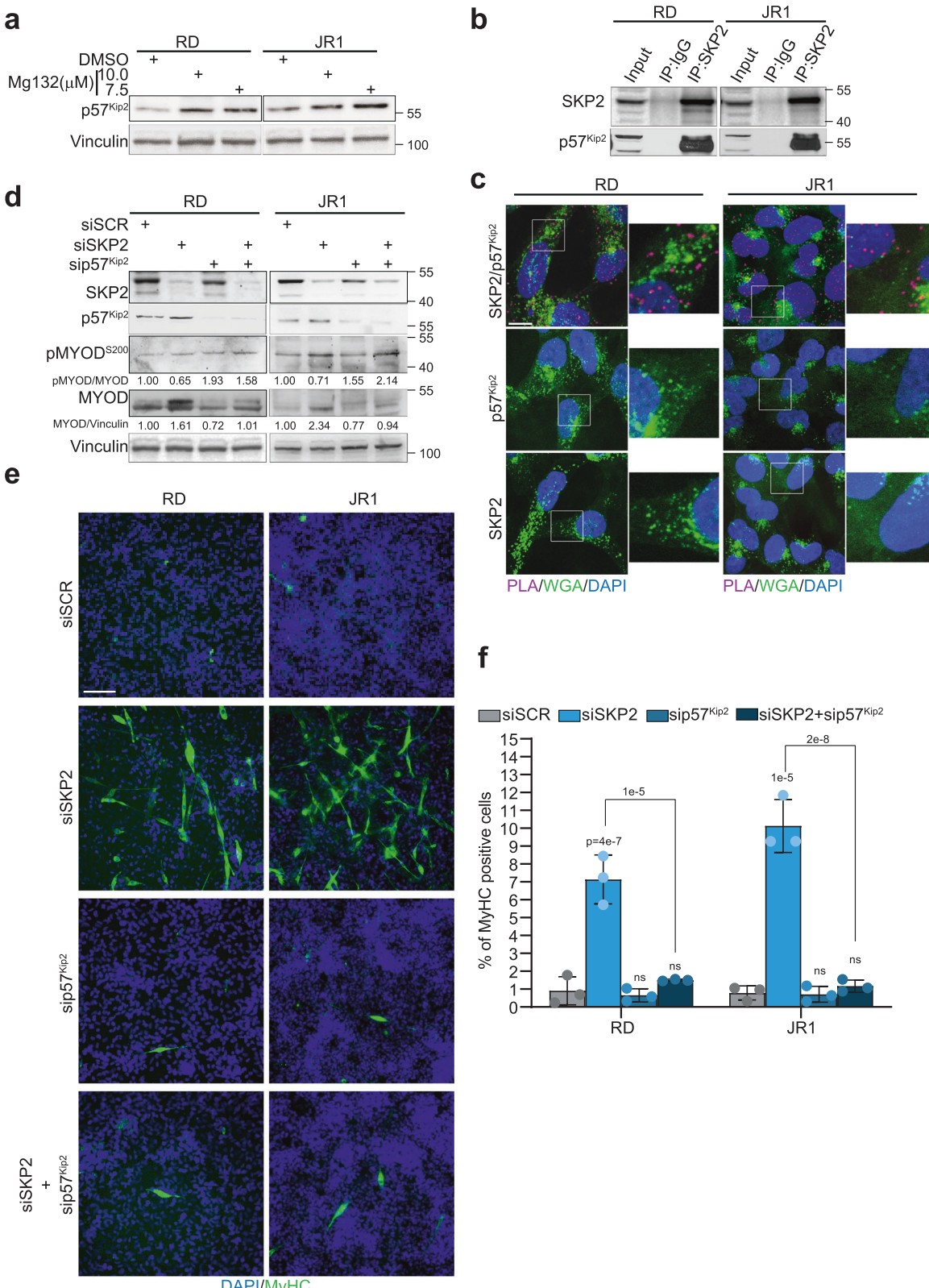

senescence in hepatocellular carcinoma[50]. Even p21[Cip1] is able to induce senescence, which is not associated to a Senescence Associated Secretory Phenotype (SASP) response activation[51], as shown in our context in Supplementary Fig. 8a. Furthermore, in the absence of Skp2, p53 loss induces senescence in normal cells in the presence of functional pRb[18], a model similar to our tumor context. Therefore, since G0/G1 cell cycle arrest is required but not sufficient for myogenesis[30],

our results suggest that when the levels of SKP2 decrease inducing the blockade of cell cycle, tumor cells can either differentiate or partly undergo to senescence.

Selective SKP2 inhibitors with different mechanisms of action have shown anti-cancer effects in vitro and in preclinical models[16,52,53]. We, here, used an in vitro validated SKP2 inhibitor, SMIP004, which degrades SKP2 with an unknown mechanism and rises p27[Kip1] levels[28]

**Fig. 5 | SKP2 depletion-induced p57$^{Kip2}$ increase is needed for myogenic differentiation. a** Representative western blot ($n = 3$ independent experiments) of the indicated proteins on RD and JR1 cells treated for 8 h with either vehicle (DMSO), 10.0 or 7.5 μM of MG132. Vinculin is the loading control. **b** Representative western blot ($n = 3$ independent experiments) of co-Immunoprecipitation of endogenous SKP2 in RD and JR1 cells showing SKP2 and p57$^{Kip2}$. **c** Representative confocal pictures ($n = 3$ independent experiments) of Proximal Ligation Assay (PLA) for SKP2 and p57$^{Kip2}$ on RD and JR1 cells. SKP2/p57$^{Kip2}$ direct close proximity/ interactions are visualized as red dots (upper). Nuclei were stained with DAPI (blue) and membrane glycoproteins with WGA (green) (see Methods). PLA negative controls were performed by using either p57$^{Kip2}$ (middle) or SKP2 (lower) antibodies

alone. Scale Bar = 10 μm. **d** Representative western blot ($n = 3$ independent experiments) of the indicated proteins on RD and JR1 cells transfected with either SCR, SKP2, p57$^{Kip2}$ or SKP2 + p57$^{Kip2}$ siRNA at 72 h post-transfection. Vinculin is the loading control. **e** Representative immunofluorescence of RD and JR1 cells treated as in (**d**) and cultured for 6 days in growth medium (GM, supplemented with 10% serum), showing expression of Myosin Heavy Chain (MyHC) (green). Nuclei were stained with DAPI (blue). Scale Bar = 100 μm. **f** Histogram depicts the quantification of MyHC-positive cells treated as in (**e**). $n = 3$ independent experiments, data presented as mean values ± SD, two-way ANOVA. Source data are provided as a Source Data file.

showing that its effects mimics those of SKP2 knockdown. However, none SKP2 inhibitor including SMIP004 have reached the clinical stage yet. A number of studies showed that SKP2 can be indirectly inhibited by the clinical stage NEDDylation inhibitor MLN4924 (Pevonedistat), which prevents the formation/activation of the SCF/CRL1 complex. MLN4924 has shown anticancer effects in preclinical adult and pediatric tumor models[17,54–57] and good tolerability in Phase I clinical trials as monotherapy and in combination studies in adult cancer patients[41,58–61]. Two Phase I studies are ongoing using this agent in combinatorial approaches also in the pediatric cancer setting involving patients with leukemia/lymphoma (NCT03349281, NCT03813147). A third pediatric study, which is the most relevant in view of our findings, is the Phase I study of the combination of MLN4924 with Irinotecan and Temozolomide in young patients with relapsed or refractory solid tumors including brain tumors (NCT03323034). The study has completed recruitment and the combination was overall well tolerated, final recommended Phase 2 dose and reports on tumor responses and overall patient outcome are pending (ASCO abstract[62]). We demonstrate here that MLN4924 mirrors the effects of SKP2 genetic suppression including PDX models. However, in contrast with SKP2 silencing and SMIP004 treatment, MLN4924 lowers MYOG levels needed for cell differentiation, in keeping with what reported on myoblasts[42], promoting Caspase3/7 activation instead. Therefore, MLN4924-induced proliferative block results in decreased survival and increased senescence rather than differentiation compared to SKP2 silencing. However, we cannot rule out effects of MNL4924 in addition to the specific SKP2-dependent effects, since it inhibits all Culling-RING ligases and can have NEDDylation-independent effects[63].

MLN4924 completely prevents in vivo growth of tumors measuring about 200 mm$^3$ and significantly slow down the growth of more advanced tumors measuring about 400 mm$^3$. Previously, a study on a panel of pediatric cancer cell lines including RMS, performed by the Pediatric Preclinical Testing Program (PPTP) group, reported intermediate activity of MLN4924 in vivo[57]. The two FN-RMS cell lines RD and JR1 used in our study, were not investigated in vivo, but RD cell line was included in the in vitro panel and showed a sensitivity comparable to our in vitro results[57].

In summary, we identify SKP2 as a pharmacologic tumor vulnerability in FN-RMS, as testified by the very low sensitivity of normal myoblasts and fibroblasts to the SKP2 inhibiting compounds, as already reported in other tumors[17,64]. However, due to the crucial role of SKP2 in cell cycle progression in normal and tumor cells, the identification of those SKP2 targeting compounds that can be more effective in that specific tumor context and the knowledge of the tumor-specific pathological mechanisms should be considered for translational purposes (refs. 53,65,66).

Collectively, our results uncover a feedback loop in which SKP2, continually induced by MYOD in FN-RMS cells, promotes cell cycle progression and prevents differentiation through the direct targeting of p27$^{Kip1}$ and p57$^{Kip2}$, thus indirectly mediating MYOD degradation (Fig. 8b). SKP2 depletion in FN-RMS causes accumulation of p21$^{Cip1}$, p27$^{Kip1}$ and p57$^{Kip2}$ resulting in cell cycle arrest and activation of a

myogenic program whereby MYOD is stabilized, up-regulates MYOG and acquire the ability to trigger terminal differentiation. The evidence that SKP2 can be pharmacologically targeted in FN-RMS through currently available NEDDylation inhibitors may have a significant translational impact considering that around 70% of high-risk FN-RMS remains fatal despite heavy multimodality therapy[1] and that SKP2 has been implicated in chemo- and radio-resistance[53,67,68]. Moreover, with this study, we provide insights into the crosstalk between epigenetic and post-translational mechanisms governing FN-RMS tumorigenesis and add evidence to the potential of a differentiation therapy in this tumor subtype.

## Methods

### Research compliances
All animal experiments were performed in accordance with the Guidelines for Animal Care and Use of the National Institutes of Health and approved by the Institutional Animal Care and Use Committee (IACUC) at the University of Texas Health Science Center, San Antonio, Texas (protocol number 20150015AR) and in accordance with the European Communities Council Directive N. 2010/63/ EU, the Italian Ministry of Health guidelines (DL 26/2014) and approved by the Italian Ministry of Health for the Children's Hospital Bambino Gesù/Plaisant animal facility of Castel Romano in Rome, Italy (protocol number 88/ 2016-PR). Human samples were handled in accordance with the ethics and principles of the Declaration of Helsinki and the approval of the study was obtained in accordance with the Institutional Review Board (IRB) of Bambino Gesù Children's Hospital, IRCCS, Rome, Italy (OPBG; Authorization 120 LB, 02/10/2015).

### Analysis of public datasets
Affymetrix profiling data from patients' public datasets and normal skeletal muscle samples were normalized and polished using the robust multiarray average in RMAexpress (Fig. 1a, b and Supplementary Fig. 1a). A representative probeset was selected for *SKP2* expression (203626_s_at) and expression levels for each patient subgroup and skeletal muscle were plotted in prism (GSE66533, GSE9103, GSE3307 and ref. 19). One-way analysis of variance (ANOVA) was used to determine any statistical significance in expression of *SKP2* between the control skeletal muscle group and each RMS subgroup. ChIP-seq data visualization was performed by IGV[69]. The analysis of the correlation between *MYOD1* and *SKP2* expression in Fig. 2b has been performed using patient dataset in ref. 19 and plotted with GraphPad Prism 8.4.3 (Dotmatics, San Diego, CA, USA)

### Clinical specimens
Sections of paraffin-embedded (FFPE) human samples were obtained from the Pathology Unit of Bambino Gesù Children's Hospital (Rome, Italy). Samples were studied in accordance with the ethics and principles of the Declaration of Helsinki and the approval of the study was obtained from the Institutional Ethical Committee of the Research Center (Authorization 120 LB, 02/10/2015). Written informed consent was obtained from all patients.

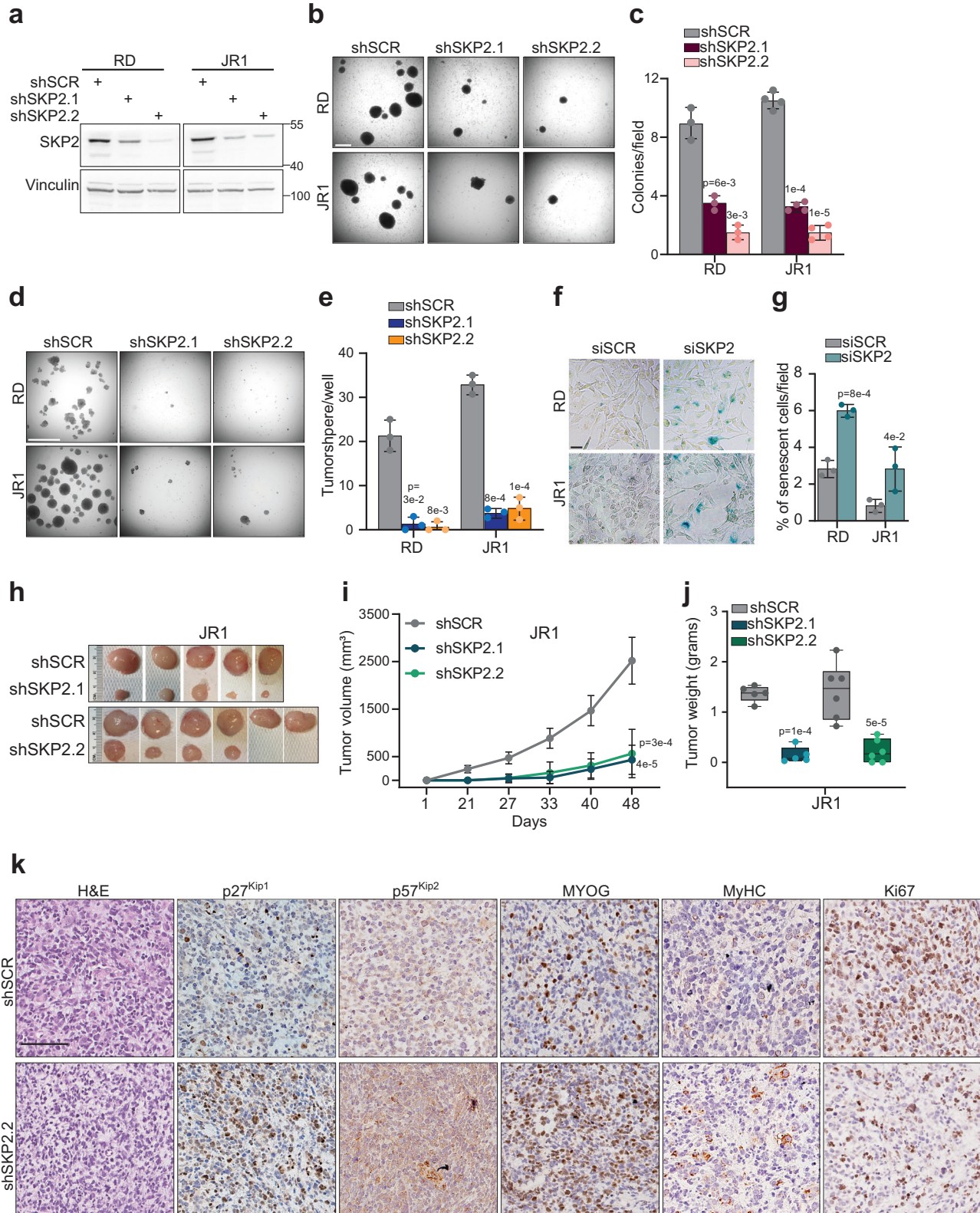

## Cell lines

RH30 (Fusion-Positive (FP) PAX3-FOXO1 RMS) and RD (Fusion-Negative (FN) RMS) cell lines were obtained from American Type Culture Collection (ATCC, Rockville, MD, USA). RD18 cells (FN-RMS), an in vitro derivative of RD cells, were a gift of C. Ponzetto (Department of Oncology, University of Turin, Turin, Italy) and were profiled for Short Tandem Repeats (STR) by BMR Genomics (Padova, Italy, http://www. bmr-genomics.it). The FN-RMS cell lines RH2, JR1 and RH36 and FP-RMS RH4 cells were provided by P. Houghton and were authenticated by STR analysis (9 loci). RH30, RD, JR1 and RH36 were primary RMS cultures established from patient-derived tumor xenografts (PDTX) collected at the St. Jude Children's Research Hospital. Several aliquots of the first culture for each cell line were stored in liquid nitrogen at −80 °C for subsequent assays. Each aliquot was passaged for a

**Fig. 6 | SKP2 silencing reduces stemness and tumorigenicity in vitro and suppresses growth in vivo. a** Representative western blot (*n* = 3 independent experiments) of the indicated proteins on RD and JR1 cells infected with either Scrambled (shSCR), SKP2.1 (shSKP2.1) or SKP2.2 (shSKP2.2) lentiviral shRNA at 72 h post-selection. Vinculin is the loading control. **b** Representative light microscopy pictures of soft agar colony formation assay on RD and JR1 cells treated as in (**a**) and grown for 2 weeks. Scale bar = 50 μm. **c** Histogram depicts the quantification of soft agar colony numbers per field. *n* = 3 (RD) and *n* = 4 (JR1) independent experiments, data presented as mean values ± SD, one-way ANOVA. **d** Representative light microscopy pictures of single cell colony formation assay on RD and JR1 cells treated as in (**a**) and grown for 2 weeks. Scale bar = 50 μm. **e** Histogram depicts the quantification of colony numbers per field. *n* = 3 independent experiments, data presented as mean values ± SD, one-way ANOVA. **f** Representative light microscopy pictures of β-Galactosidase staining of RD and JR1 cells transfected with either SCR or SKP2 siRNA. Scale bar = 100 μm. **g** Histogram depicts the quantification of the percentage of senescent cells per field. *n* = 3 independent experiments, data presented as mean values ± SD, Student's two-tailed *t*-test. **h** Images of JR1 shSCR, shSKP2.1, and shSKP2.2 tumors explanted from mice post euthanasia at 48 days post-inoculation. **i** Tumor volume of shSCR (*n* = 11), shSKP2.1 (*n* = 5) and shSKP2.2 (*n* = 6) JR1 xenografts assessed by caliper measurement represented in mm³ followed for 48 days post-inoculation. Data presented as mean values ± SD, two-way ANOVA. **j** Tumor weight of shSCR (*n* = 11), shSKP2.1 (*n* = 5) and shSKP2.2 (*n* = 6) JR1 xenografts. Box plots show 25th to 75th quartiles, black bar shows the median, and whiskers go down to the smallest value and up to the largest. One-way ANOVA **k** Representative images (*n* = 3 independent experiments) of H&E, p27$^{Kip1}$, p57$^{Kip2}$, MYOG, MyHC, and Ki67 immunohistochemistry of tumor sections from JR1 xenografts expressing either shSCR or shSKP2.2. Scale Bars = 100 μm. Source data are provided as a Source Data file.

maximum of 4 months. RH30, RH36, RH4, RH2 and RH4 Cas9 cells were cultured in RPMI 1640 (Invitrogen, Carlsbad, CA, USA), RD, JR1, RD18 cells were cultured in DMEM high-glucose (Invitrogen, Carlsbad, CA, USA) both supplemented with 10% fetal bovine serum (FBS), 1% L-glutamine and 1% penicillin-streptomycin. HSMM (CC-2580, Lot 0000424745) were purchased from Lonza (Walkersville, MD, USA). Vendors also supplied growth media with supplements and serum for HSMM cells (CC-3245). HSMM differentiation medium was prepared by adding 2% horse serum to DMEM-F12 medium (both from Lonza, Walkersville, MD, USA).

C2C12 (CRL-1772) were purchased from ATCC (Rockville, MD, USA) and cultured in DMEM high-glucose (Invitrogen, Carlsbad, CA, USA) supplemented with 10% fetal bovine serum (FBS), 1% L-glutamine and 1% penicillin-streptomycin. C2C12 differentiation medium was prepared by adding 2% horse serum to DMEM medium.

C3H/10T1/2 (Clone 8 catalog CCL-226) were purchased from ATCC (Rockville, MD, USA) and cultured in Eagle's Basal medium (21010-046, Thermo Fisher Scientific, Lafayette, CO, USA) supplemented with 10% fetal bovine serum (FBS), 1% L-glutamine and 1% penicillin-streptomycin. Normal Human Lung Fibroblast (NHLF) were purchased from LONZA (Lonza, Walkersville, MD, USA). Vendors also supplied growth media with supplements and serum (CC-3132). Orthotopic patient-derived xenograft (o-PDX) cells (p53-mutated SJRHB011_YC, p53 wild-type (from two different tumor sites) SJRHB012_YC and SJRHB012_ZC, here reported as B011_YC, B012_YC, and B012_ZC, respectively) were obtained through the Childhood Solid Tumor Network (CSTN) at St. Jude Children's Hospital (Memphis, TN, USA)[43]. They were cultured on dishes coated with Gelatin (G9391, Sigma-Aldrich, St Louis, MO, USA) in NB medium (21103049, Thermofisher Scientific, Lafayette, CO, USA), supplemented with 2x B-27 (17504044, Thermofisher Scientific, Lafayette, CO, USA), 100 U/ml penicillin /streptomycin and 2 mM Glutamax. Medium was replaced with fresh one three times per week. When reaching confluency, cells were detached using Accutase (A6964, Sigma-Aldrich, St Louis, MO, USA) and splitted in a ratio of 1:2 to 1:3. For Gelatin-coating, a 2% solution of Gelatin in water was left on the dish for 2 h at 37 °C. The solution was then removed, and the plates were dried at RT for 30–60 min. All the cell lines were cultured at 37 °C in a humidified atmosphere of 5% CO$_2$/ 95% air and regularly checked for mycoplasma contamination.

**Transient RNA interference**
Cells were transfected as described in[70] with siRNAs against either human *SKP2* (SASI_Hs02_00340672s), mouse *SKP2* (SASI_Mm02_00322740), human *MYOD* (MYOD.1: CUUUGCCACAACGGACGACUU; MYOD.2: ACUUCUAUGACGACCCGUGUU; MYOD.3: GCUACGA-CACCGCCUACUACA), human p27$^{Kip1}$/*CDKN2B* (custom: ACGUAAA-CAGCUCGAAUUA), human p57$^{Kip2}$/*CDKN2C* (custom: GAACCGG CUGGGAUUACGACUU) or with a non-targeting siRNA as control (SIC001) (Sigma-Aldrich, St Louis, MO, USA) (100 nM final

concentration) using Oligofectamine (Invitrogen, Carlsbad, CA), according to the manufacturer's recommendations. Twenty-four hours later, the medium was replaced with fresh growth medium supplemented with 10% FBS, 1% L-glutamine and 1% penicillin-streptomycin, and the transfected cells were harvested at different time points. ON-TARGETplus SMART pool siRNA against human *SKP2* (L-003324-00-005) or non-targeting control siRNA (D-001810- 10-05) (Dharmacon, Thermo Fisher Scientific, Lafayette, CO, USA) were used.

**Lentivirus production and cell infection**
RD and JR1 cells were infected with lentiviral pLKO.1-puro vectors individually expressing two different shRNA sequences (TRCN0000007532 or TRCN0000007534 from RNAi consortium; Sigma-Aldrich, St Louis, MO, USA) used to target human *SKP2*. A lentiviral vector expressing a non-targeting shRNA sequence was used as control (SHC002; Sigma-Aldrich, St Louis, MO, USA). RD and RH4 stably expressing Cas9 cells[20] were co-infected with two different LRG2_1T lentiviral vectors expressing specific sgRNA sequences for MYOD1 (CCGCTATATCGAGGGCCTGC and CAGGCCCTCGATA-TAGCGGA). A lentiviral LRG2_1T vector expressing a non-targeting sgRNA sequence was used as control. RD and JR1 cells were infected with lentiviral Lenti-ORF clone of CDKN1C (RC209840L3, Origene, Rockville, MD, USA). Lentiviruses were produced in HEK293T packaging cells seeded in 100 mm culture dishes and transfected using Fugene 6 (Promega, Madison, WI, USA) with lentiviral packaging vectors psPAX2 (0.44 μg; Addgene) and pCMV-VSV-G (0.87 μg; Addgene) and 2.6 μg of each specific shRNA or sgRNA, in 10 ml of DMEM medium (Lonza Group Ltd, Basel, Switzerland) supplemented with 10% FBS, 1% glutamine and 1% penicillin-streptomycin. Transfection medium was replaced 24 h later with new complete DMEM and 48 h after transfection the lentiviral containing medium was collected, spun to remove cell debris, and the supernatant filtered through a 0.45 μm low protein binding filters. The viral titer was calculated using the p24 ELISA (Cell BIOLABS, Inc., San Diego, CA, USA) and viral aliquots immediately stored at −80 °C. For lentiviral infection, RD and JR1 cells were plated in a 100 mm dish (1.2 × 10⁶ cells) and, 24 h later, transduced at a 5–10 multiplicity of infection (MOI) for 24 h in the presence of polybrene (8 μg/ml; Sigma-Aldrich, St Louis, MO, USA) and 10% FBS. After a further 24 h, RD and JR1 cells were selected with 1 μg/ml and 1.5 μg/ml puromycin for 3 days, respectively. Cells were harvested at different time points for subsequent experiments.

**Luciferase activity assay**
1Kb from the *SKP2* enhancer region was amplified from human genomic DNA using specific primer (FW: CCGGGCTAGCCCGGAGT CTTTGCTATTTCC, REV: CCGGCTCGAGCAGAATCTCAGGAATTTTAC) and cloned in PGL3-promoter vector (E1761 Promega, Madison, WI, USA). For luciferase assays, 1×10⁵ HEK293T cells were transfected with 100 ng of PGL3-promoter-SKP2enhancer or PGL3-promoter-EV reporter vector, 2 ng of pRL-CMV-Renilla luciferase vector (Promega) and

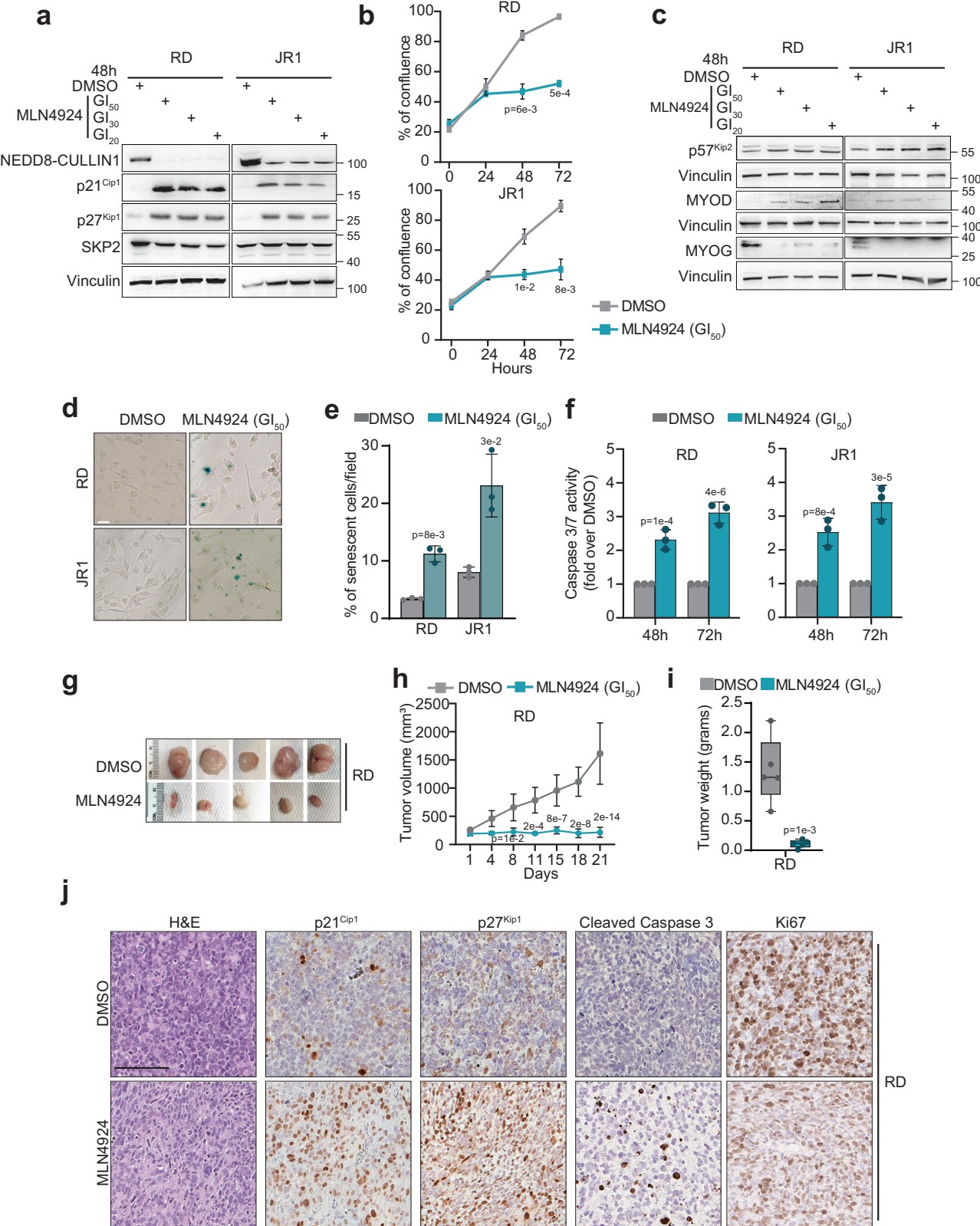

300 ng of either FLAG-hMYOD1 (#794 Addgene), or empty vector (as a control) using Fugene 6 (Promega, Madison, WI, USA) according to the manufacturer's protocol. Luciferase activity was measured 24 h after transfection using the Dual Luciferase Reporter Assay System (Promega, Madison, WI, USA) and light emission was measured over 10 seconds using EnSpire Multimode Plate Reader (PerkinElmer,

Waltham, MA, USA). The transfection efficiency was normalized to Renilla luciferase activity.

**Retrovirus production and cell infections**

To obtain the MSCV-GFP-SKP2 retroviral vector, a cDNA encoding *SKP2* (kindly provided by B. Shultz in pGEX vector) was subcloned into the

**Fig. 7 | NEDDylation inhibition prevents SKP2 functions and suppresses tumor growth in vivo. a** Representative western blot ($n = 3$ independent experiments) of the indicated proteins on RD and JR1 cells treated for 48 h with either vehicle (DMSO) or MLN4924, at the reported Growth Inhibition (GI) concentrations (refer to Fig. S7A for details). Vinculin is the loading control. **b** Growth curve analysis of RD and JR1 cells treated as in (**a**). $n = 3$ independent experiments, data presented as mean values ± SD, Student's two-tailed $t$-test. **c** Representative western blot ($n = 3$ independent experiments) of the indicated proteins on RD and JR1 cells treated as in (**a**). Vinculin is the loading control. **d** Representative light microscopy pictures of β-Galactosidase staining (blue) of RD and JR1 cells treated with vehicle (DMSO) or $GI_{50}$ MLN4924. Scale bar = 100 μm. **e** Histogram depicts the quantification of the percentage of senescent cells per field of RD and JR1 cells treated as in (**d**). $n = 3$ independent experiments, data presented as mean values ± SD, Student's two-tailed $t$-test. **f** Histogram depicts the quantification of Caspase 3/7 activity at the reported MLN4924 concentration and time points calculated as fold increase over vehicle (DMSO) ($n = 3$ independent experiments, data presented as mean values ± SD, two-way ANOVA. **g** Images of RD tumors explanted from mice treated with either vehicle or MLN4924 for 3 weeks. **h** Tumor volume of RD xenografts treated as in (**g**) (vehicle $n = 5$, MLN4924 $n = 5$) assessed by caliper measurement represented in mm³ followed for 3 weeks treatment. Data presented as mean values ± SD, two-way ANOVA. **i** Tumor weight of transplanted RD xenografts treated as in (**g**). Box plots show 25th to 75th quartiles, black bar shows the median, and whiskers go down to the smallest value and up to the largest. One-way ANOVA. **j** Representative images ($n = 3$ independent experiments) of H&E, p27$^{Kip1}$, p21$^{Cip1}$, Cleaved Caspase and Ki67 immunohistochemistry of tumor sections from RD xenografts treated with either vehicle or MLN4924. Scale Bars = 100 μm. Source data are provided as a Source Data file.

retroviral vector MSCV-GFP. Briefly, *SKP2* cDNA was extracted from pGEX using EcoRI and NotI restriction enzymes, MSCV-GFP retroviral vector was linearized with XhoI and both fragments were treated to generate blunt-ends for further ligation. The control vector was the MSCV-GFP empty vector. C3H/10T1/2 murine fibroblasts were transduced with either a pBABE retro vector expressing murine MyoD (Plasmid #20917, Addgene) or a pBABE empty vector as control. To produce the retroviral particles, HEK293GP cells (kindly provided by G.M. Fimia, National Institute for Infectious Diseases, I.R.C.C.S. Lazzaro Spallanzani, Rome, Italy) were cultured in DMEM supplemented with 10% FBS, 1% L-glutamine and 1% penicillin-streptomycin and transiently transfected using ProFection® Mammalian Transfection System (E1200, Promega, Madison, WI, USA). Supernatant containing viral particles collected after 72 h was used to infect cells O/N in the presence of polybrene (8 μg/ml). Cells were harvested 48 h after infection for subsequent experiments.

### Drugs
MG132 (HY-13259) and MLN4924 (Pevonedistat, HY-70062) were purchased from MedChemExpress (Monmouth Junction, NJ, USA). SMIP004 (143360-00-3) was purchased from Sigma-Aldrich (St Louis, MO, USA). Compounds were dissolved in DMSO at 10 mmol/L.

### Determination of the GI50 and cell proliferation assay
Growth Inhibition (GI)$_{50}$, $GI_{30}$ and $GI_{20}$ were determined as previously described in ref. 71. Briefly, RD, JR1 and PDX cells were seeded on 384-well plates in previously described media. After 24 h were treated with decreasing doses of MLN4924 (9 μM–0.05 nM), or with DMSO. Cells were plated to achieve 20% confluence at the time of drug treatment ($1.2 \times 10^3$/well) and monitored until 72 h, when control (DMSO-treated) wells reached ~90% confluence. GI50, GI30, and GI20 values were calculated 72 h post treatment using the GraphPad™ Prism version 8. For proliferation experiments, were seeded on 96-well plates (3000 cells/well) and, after 24 h (t0), new media containing DMSO or the drugs at the selected concentrations were added to the wells. The percentage of cell confluence was quantified under phase contrast every 24 h using the Celigo Image Cytometer (Nexcelom Bioscience, Lawrence, MA, USA). For PDX cell viability measurement, was used the Cell Titer-Glo Assay (G7570, Promega, Madison, WI, USA) according to the protocol from the manufacturer.

### Determination of Caspase 3/7 activity
RD and JR1 cells were seeded into 96-well, black, flat bottom plates at a concentration of 5000 cells per well. For evaluation of apoptosis effects, after 24 h cells were treated with MLN4924 at the indicated doses in triplicates. The measurement of Caspase-3/7 activity has been determined using Caspase-Glo-3/7 (G8090, Promega, Madison, WI, USA), according to the manufacturer's instructions. The activity of Caspase-3/7 was then examined using EnSpire Multimode Plate Reader (PerkinElmer, Waltham, MA, USA).

### Real-time RT-quantitative PCR
Total RNA was extracted using TRIzol (Invitrogen, Carlsbad, CA, USA) according to the manufacturer's protocol and inspected by agarose gel electrophoresis. Reverse transcription was performed using the Improm-II Reverse Transcription System (A3800, Promega, Madison, WI, USA). The expression levels were measured by qRT-PCR for the relative quantification of the gene expression. TaqMan gene assay (Applied Biosystems, Life Technologies, Carlsbad, CA, USA) for human *SKP2* (Hs01021864_m1) murine *Skp2* (Mm00449925_m1), human *MYOD1* (Hs02330075_g1), murine *MyoD1* (Mm00440387_m1), *MYOG* (Hs01072232_m1), *MYH2* (Hs00430042_m1), p21$^{Cip1}$ (*CDKN1A*) (Hs00355782_m1), p27$^{Kip1}$ (*CDKN1B*) (Hs01597588_m1), *MEF2D* (Hs00954735_m1) and p57$^{Kip2}$ (*CDKN1C*) (Hs00175938_m1), were used. Values were normalized according to the human glyceraldehyde-3-phosphate dehydrogenase (*GAPDH*) (Hs99999905_m1) and murine *Hprt* (Mm1545399_m1) mRNA levels. An Applied Biosystems 7900HT Fast RealTime PCR System (Applied Biosystems, Waltham, MA, USA) was used for the measurements. The expression fold change was calculated by the 2- ΔΔCt method for each of the reference genes. At least three independent amplifications were performed for each probe, in triplicate.

### Western blotting
Western blotting was performed on whole-cell lysates by homogenizing cells in RIPA lysis buffer (50 mM Tris pH 7.4, 150 mM NaCl, 1% Triton X-100, 1 mM EDTA, 1% sodium deoxycholate, 0.1% SDS), containing the protease inhibitor cocktail (Sigma-Aldrich, St Louis, MO, USA), NaF 1 mM, Na$_3$VO$_4$ 1 mM and PMSF 1 mM. Lysates were incubated on ice for 30 min and centrifuged at 12.000 × g for 20 min at 4 °C. Supernatants were then quantified with BCA Protein Assay Kit (Pierce, Life Technologies, Carlsbad, CA, USA) according to the manufacturer's protocol and then boiled in reducing SDS sample buffer (200 mM Tris–HCl pH 6.8, 40% glycerol, 20% β-mercaptoethanol, 4% sodium dodecyl sulfate, and bromophenol blue); and 30 μg of protein lysate per lane was run through either 10% or 12% SDS-PAGE gels, and then transferred to Hybond ECL membranes (Amersham, GE HEALTHCARE BioScience Corporate Piscataway, NJ, USA). Membranes were blocked for 1 h in 5% non-fat dried milk in Tris-buffered saline (TBS) and incubated overnight with the appropriate primary antibody at 4 °C. Membranes were then washed in TBS and incubated with the appropriate secondary antibody. Both primary and secondary antibodies were diluted in 5 % non-fat dried milk in TBS. Membranes were then incubated with HRP-conjugated secondary antibody for 1 h at room temperature. Detection was performed by ECL Western Blotting Detection Reagents (Amersham, GE HEALTHCARE BioScience Corporate Piscataway, NJ, USA). Antibodies against SKP2 (H-435, sc-7164, 1:500), p27$^{Kip1}$ (F-8, sc-1641, 1:500), MYOD (5–8 A, sc-32758, 1:200), pMYOD (Ser-200) (sc-101741, 1:200) and β- ACTIN (C4, sc-47778, 1:1000) were from Santa Cruz Biotechnology Inc., (Santa Cruz, CA, USA); p21$^{Cip1}$ (12D1, #2947, 1:1000), Nedd8 (19E3, #2754, 1:1000) and

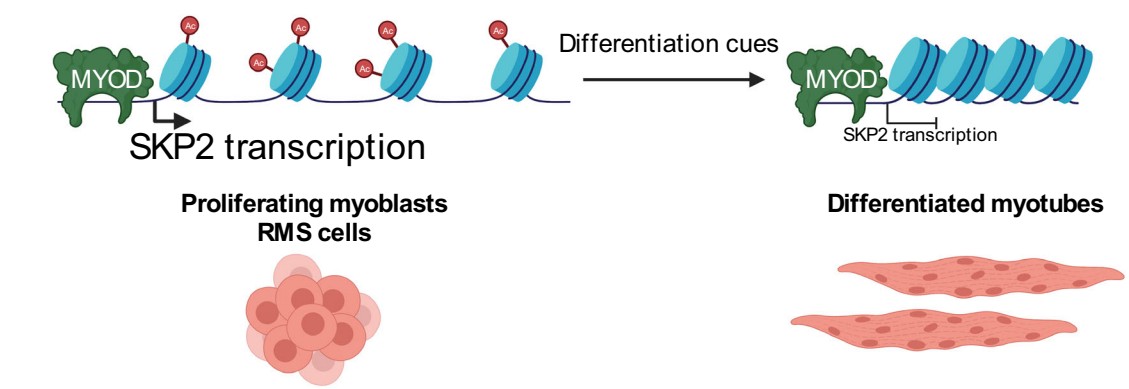

**a**

MYOD — SKP2 transcription

**Proliferating myoblasts**
**RMS cells**

Differentiation cues →

MYOD — SKP2 transcription

**Differentiated myotubes**

**b**

*Fig. 8 | SKP2 effects are mediated by p27^Kip1 and p57^Kip2 degradation. a* A model showing that in myoblasts and RMS cells that are both proliferating, H3K27ac (Ac) enrichment at *SKP2* DNA regulatory regions allows MYOD, which is bound on these regions, to induce the expression of *SKP2* thus supporting proliferation (left). Upon differentiation cues, reduction of H3K27ac enrichment results in a closed chromatin conformation, impairing MYOD ability to induce *SKP2* transcription. These events promote differentiation, which is inhibited in RMS (right). *b* A model showing that in RMS cells, MYOD induces the expression of *SKP2*, which in turn associates with the SCF/CRL1 complex to promote ubiquitylation and proteasomal degradation of p27^Kip1 and p57^Kip2 by direct interaction and p21^Cip1 in an indirect way, promoting oncogenic features (left). These SKP2 functions can be suppressed by SKP2 silencing (upper right) or by its functional inhibition by the SKP2 inhibitor SMIP004 or the NEDDylation inhibitor MLN4924 (lower right) to block RMS oncogenic properties. NAE NEDD8-activating enzyme, Ub Ubiquitin, N8 NEDD8. Created with BioRender.com.

GAPDH (D16H11, #5174, 1:2000) were from Cell Signaling Technology, Inc., (Danvers, MA, USA); MYOG (F5D-c, 1:200) was from DSHB (University of Iowa, Iowa City, IA, USA); α-TUBULIN (NB100-680, 1:5000) was from Novus Biologicals (Littleton, CO, USA); VINCULIN (hVIN-1; V9131, 1:5000) was from Sigma (Sigma-Aldrich, St Louis, MO, USA); p57[Kip2] (#556346, 1:500) were from BD Biosciences (San Jose', CA, USA). All secondary antibodies (1:5000) were obtained from Cell Signaling Technology, Inc., (Danvers, MA, USA). All the antibodies were used in accordance with the manufacturer's instructions. All uncropped versions of western blots are reported in the Source Data file and in the Supplementary Information file.

### Soft-agar colony formation assay

Lentiviral infected RD and JR1 cells were assayed for their capacity to form colonies in soft-agar 3 days after puromycin selection, as described. A total of $10^4$ cells were suspended in DMEM or RPMI (10% FBS) with 0.35% Agar (50081 NuSieve GTG Agarose, Lonza, Walkersville, MD, USA). Cells were plated on a of 0.7% Agar layer in DMEM or RPMI (10% FBS) in 6 multi-well plates. Every 2 days medium was refreshed. On week 2 colonies were counted by microscopic inspection and images were acquired with an Olympus Digital Camera XC50 (Olympus Corporation, Shinjuku-ku, Tokyo, Japan). Duplicate assays were carried out in three independent experiments.

### Sphere formation assay

Twenty-five x$10^3$ lentiviral infected RD and JR1 cells were resuspended in 2000 μL Neurobasal Medium (21103049, ThermoFisher Scientific, Rockford, USA) supplemented with 2X B27 (17504044, ThermoFisher Scientific, Rockford, USA), 1% penicillin-streptomycin, 20 ng/mL EGF (AF-100-15) and 20 ng/mL bFGF (AF-100-18B) both from PeproTech (London, UK). Cells were plated in a Low Attachment 6-well plate (3471, Corning, New York, USA). The cells were incubated for at least 10–12 days with fresh media added every 3 days.

### Cell cycle assay

After siRNA transfection, RD, JR1, and RD18 cells were analyzed by flow cytometry as reported[72]. Briefly, cells were harvested by trypsinization 48 h after transfection, washed in cold phosphate buffered saline (PBS), fixed in cold 50% PBS and 50% acetone/methanol (1:4 v/v) for at least 1 h. Fixed cells were pelleted 5 min 1500 rpm and alcoholic fixative removed. Pellet was stained in the dark with a solution of 50 μg/ml propidium iodide (PI) (ThermoFisher Scientific, Rockford, USA) and 50 μg/ml RNase (Sigma-Aldrich, St Louis, MO, USA) for 30 min at room temperature. Stained cells were analyzed for cell cycle by fluorescence-activated cell sorting using a FACSCantoII equipped with a FACSDiva 6.1 CellQuestTM software (Becton Dickinson Instrument, San Josè, CA, USA). The percentage of cells in G0/G1, S, and G2/M phases was expressed as relative change compared to Scrambled siRNA transfected cells. The gating strategy has been reported in Fig. S12.

### Senescence β-galactosidase staining assay

RD and JR1 cells were either transfected with siRNAs or treated with vehicle/MLN4924 for 72 h, and then fixed with PFA 4% for 15 min. β-galactosidase staining was performed using the Senescence β-Galactosidase Staining Kit (9860, Cell Signaling Technology, Inc., Danvers, MA, USA) according to the manufacturer instructions. Images were acquired with a Leica microscope (Leica Microsystems, Mannhein, Germany).

### Immunofluorescence

Cultured cells were fixed with 2% PFA for 10 min, permeabilized with 0.5% Triton/PBS, and blocked with 4% BSA in PBS 1 h at room temperature, as reported in ref. 72. Immunostaining with anti-MyHC (MF-20, 1:100; DSHB, University of Iowa, Iowa City, IA, USA) was performed 1 h at room temperature. Antibody binding was revealed using species-specific secondary antibodies coupled to Alexa Fluor 488 (1:1000 Invitrogen, A11017) or 555 (1:1000 Invitrogen, A21425). Nuclei were visualized by counterstaining with DAPI. Images were acquired with a Leica microscope (Leica microsystems, Mannhein, Germany).

### Protein Co-Immunoprecipitation

RD and JR1 cells were lysed in Triton buffer (50 mM Tris-HCl pH 7.5, 250 mM NaCl, 50 mM NaF, 1 mM EDTA 1 pH 8, 0.1% Triton) supplemented with proteases and phosphatases inhibitors. After preclearing for 1 h at 4 °C, immunoprecipitation was performed by incubating 1.5 mg of whole-cell protein extracts with 1.5 ug of either anti-SKP2 mouse (32-3300, Thermofisher Scientific, Lafayette, CO, USA, 1:1000 w-w), anti-p21[Cip1] (12D1, #2947, Cell Signaling Technology, Inc., Danvers, MA, USA, 1:1000 w-w) and with anti-IgG antibody at the same concentration, with rocking at 4 °C overnight. The immune complexes were collected by incubation with Dynabeads protein A or G (10003D, 10001D, Invitrogen, Carlsbad, CA, USA) for 1 h and washed with Triton buffer. The beads were, then, resuspended in 25 μl SDS Laemmli sample buffer, subjected to SDS-PAGE (10% polyacrylamide) analysis, and electro-transferred onto PVDF membranes. The membranes were probed with primary antibodies as described above.

### In situ proximity ligation assay (PLA)

A DuoLink PLA kit (DUO92101, Sigma-Aldrich, St Louis, MO, USA) was used to detect protein–protein interactions as per manufacturer's protocol. RD and JR1 cells were seeded on Nunc Lab-Tek II Chamber Slide™ System (154534, Thermofisher Scientific, Lafayette, CO, USA). Antibodies against anti-SKP2 rabbit (H-435, sc-7164, Santa Cruz, CA, USA, 1:25), anti-SKP2 mouse (32-3300, Thermofisher Scientific, Lafayette, CO, USA, 1:50), anti-p21[Cip1] (12D1, #2947, Cell Signaling Technology, Inc., Danvers, MA, USA, 1:500), anti-p27[Kip1] (C-19, sc-1641, Santa Cruz, CA, USA, 1:25), anti-p57[Kip2](#556346, BD Biosciences, San Jose, CA, USA, 1:25). Nuclei were visualized by counterstaining with DAPI and membrane glycoproteins with Wheat Germ Agglutinin (WGA). The signal was detected as a distinct fluorescent dot in the Texas red channel and analyzed by fluorescence Olympus FV3000 confocal microscopy with Olympus FV315S-SW image acquisition software.

### Chromosome conformation capture (3 C)

RD and RH4 cells ($1 \times 10^7$ per sample) were incubated in DMEM containing 1% or 0.5% formaldehyde for 10 min, then glycine was added to a final concentration of 0.125 mol/L, and the cells were lysed using a lysis buffer (50 mM Tris, 150 mM NaCl, 5 mM EDTA, 1% TX-100 and protease inhibitors) for 10 min on ice. A not cross-linked sample for each cell line was used as a technical control for the next steps. Nuclei were collected by centrifugation and resuspended in $H_2O$ and restriction enzyme (Dpn II) buffer, then SDS to a final concentration of 0,3% was added to the samples and the nuclei were incubated for 1 h at 37 °C while shaking at 900 RPM. Afterward, Triton X-100 was added to a final concentration of 3%, and the samples were incubated for 1 h at 37 °C while shaking at 900 RPM to sequester the SDS. Cross-linked DNA was digested overnight with 600 units DpnII. Samples were then diluted in T4 DNA ligase buffer, T4 DNA ligase (50 units/sample) was added and incubated overnight at 16 °C. After adding proteinase K (10 mg/mL), crosslinks were reversed by overnight incubation at 65 °C followed by an incubation with RNaseA (10 mg/mL) for 45 min at 37 °C. DNA was extracted in phenol/ chloroform and precipitated in ethanol.

In order to obtain a negative control for the 3 C analysis, in addition to the non–cross-linked sample, we selected a fragment chr5:36125–36138 that did not show any characteristic that can be associated with a regulatory region and for which is not expected an interaction with the promoter region. *SNAI2* enhancer region[6] has been used as positive control. PCR products were resolved on 2% agarose gels. In order to normalize 3C-PCR signals, we used a loading control (internal primers located in the *GAPDH* gene). The amount of DNA

input was first titrated, and bands analyzed semiquantitatively using ImageJ software; the background was subtracted, and data normalized to an internal region unaffected by the restriction digest (LC region) (21,22). Two biological replicates were prepared and analyzed in three technical repeats. The primers used to generate each fragment are: P1 (ACGGGTAAAGCTCGTTGCAA) and P2 (TATACCAGCTGTCCCTCATC) for the intronic *SKP2* enhancer; P3 (CGTATGGGCTATTTAGGTGT) and P4 (ATCCATTGGGCAAACTATAC) for negative control; P5 (GAATTG-TAGGGGGAACAGAC) and P6 (GGCCGCGTGCAAATTAAGTA) for *SNAI2* enhancer positive control; LC FW (ACGTGGACTTACAACAGAGA) and LC REV (GCAGCCCAGAACCAGTCATC) for loading control.

## RNA-seq sample preparation and data analysis

RNA was extracted from JR1 and RD cells previously infected with lentiviruses expressing either shRNA sequence against *SKP2* or scrambled non-targeting (shSCR), using the RNeasy mini kit (#74104, Qiagen, Hilden, Germany) according to manufacturer's instruction. Poly-A selected RNA libraries were prepared using the NEBNext Ultra II Directional RNA Library Prep kit and sequenced on an Illumina Nova-Seq6000 System (2x150bp). Reads were aligned to the GRCh38 reference genome using STAR version 2.7.9a, and gene expression was calculated as Transcript Per Million (TPM) reads using RSEM version 1.3.2 via a maximum likelihood estimation framework. Gene Set Enrichment Analysis (GSEA) was performed using ranklists of log2-fold change in TPM, comparing each shSKP2 condition to its paired shSCR control RNA-seq experiment. Bubble-plots of enriched output from GSEA analysis were created in R using custom scripts.

## ChIP-seq sample preparation and data analysis

Formaldehyde-fixed (1%, 10 min) cells (RD and JR1) were digested to achieve chromatin fragmented to a range of 150–900 bp using Micrococcal Nuclease. Chromatin samples were immunoprecipitated overnight at 4 °C with antibodies targeting MYOD (#13812, Cell Signaling Technology, Inc., Danvers, MA, USA, 1:10 w:w), and H3K27ac (#39133, Active Motif, Carlsbad, CA, USA, 1:1 w-w). DNA purifications were performed with the SimpleChIP Enzymatic Chromatin IP Kit (#9003, Cell Signaling Technology, Inc., Danvers, MA, USA). We employed ChIP-seq spike in using Drosophila chromatin (#53083, Active Motif, Carlsbad, CA, USA) and Drosophila-specific histone variant H2Av antibody (#61686, Active Motif, Carlsbad, CA, USA, 1:2 w:w). ChIP-seq libraries were prepared using the NEBNext Ultra II DNA Library Prep kit and sequenced on an Illumina NovaSeq6000 (2x150bp). Reads were aligned to the GRCh38 reference genome using BWA version 0.7.17 and samples were normalized to reads per million mapped *Drosophila* spike-in reads, previously aligned on dm3 reference genome. Peak calling was performed using MACS3 and mapping artifacts were removed by excluding regions listed in ENCODE exclusion list (ENCFF356LFX). Heatmaps and average density profiles were generated using deepTools by plotting H3K27ac and MYOD ChIP-seq read coverages over MYOD-bound genomic regions.

## HiChIP sample preparation and data analysis

HiChIP was performed using the Dovetail MNase-HiChIP kit (#21007, Dovetail Genomics, Scotts Valley, CA, USA) following manufacturer's instruction, with few changes. Briefly, $10 \times 10^6$ RD FN-RMS and RH4 FP-RMS cells were collected and snap-frozen at −80C for at least 30 min. Cells were then crosslinked with disuccinimidyl glutarate (DSG, #A35392, Thermo Fisher, Lafayette, CO, USA) and 37% formaldehyde. Chromatin lysates extracted from C2C12 mouse cells were used as spike-in for AQuA-normalization and chromatin immunoprecipitation of H3K27ac (MABE647, Millipore, Burlington, MA, USA) was performed overnight. The following day, high-salt washes were performed after protein A/G beads 1 h incubation and pull-down. Specifically, samples were washed 2x with RIPA, 2x with high-salt buffer (500 mM NaCl), 2x

with LiCl wash buffer (250 mM LiCl) and 2x with Dovetail wash buffer (150 mM NaCl). Proximity-ligation and library preparation were performed according to the manufacturer's instructions and samples were sequenced on an Illumina NovaSeq System (2x150bp). To obtain valid read pairs, paired-end reads were mapped to the human (hg38) or mouse (mm10) reference genome using Bowtie2 within the HiC-Pro pipeline[73]. Reads were then filtered for read pairs with contact range >1000 bp and final read pairs were visualized on Juicebox[74]. All downstream analyses requiring sub-matrix extraction from.hic files were performed using Juicer and strawr[75]. Contact maps and Aggregated Peak Analysis (APA) plots were used to visualize 3D contacts.

## Immunohistochemistry

Immunohistochemistry was performed on 2 µm-thick sections obtained from formalin-fixed tissue embedded in paraffin. After dewaxing and rehydrating, heat-induced epitope retrieval was performed by boiling the slides with EDTA (pH 9) (Dako, Glostrup, Denmark). Endogenous peroxidase was blocked with 3% hydrogen peroxide followed by incubation with mouse-to-mouse blocking reagent to inhibit endogenous mouse immunoglobulin and then another blocking with blocked BSA 5%. Sections were incubated overnight at +4 °C with mouse monoclonal α-SKP2 antibody (dilution 1:50; 32-3300, Thermofisher Scientific, Lafayette, CO, USA), mouse monoclonal α-p27$^{Kip1}$ antibody (dilution 1:50; SX53G8.5, sc-53871, Santa Cruz); mouse monoclonal α-p57$^{Kip2}$ antibody (dilution 1:100; KP39, sc-56341, Santa Cruz); mouse monoclonal α-MYOG antibody (dilution 1:100; F5D-c, DSHB); Rabbit polyclonal α-Cleaved Caspase 3 antibody (dilution 1:300; #9661, Cell Signaling Technology); Rabbit polyclonal α- p21$^{Cip1}$ (1:50, 12D1, Cell Signaling Technology Cat# 2947); or mouse monoclonal α-MyHC antibody (dilution 1:50; MF20, DSHB) and α-Ki67 antibody (ready to use; IR626, Dako). Detection of the primary antibody was performed by using the appropriate secondary biotinylated antibody and the peroxidase DAB kit with or without counterstaining with Gill's hematoxylin (Dako, Carpinteria, USA). Negative controls were stained in parallel with only the primary antibody. Histological image acquisition and analysis have been performed by a technician and a pathologist in a blinding manner. The light microscopy imaging was performed on a Nikon E600 light microscope equipped with NIS Elements BR software, using 20x objective.

## In vivo xenograft models

NOD.Cg-Prkdc$^{scid}$ Il2rg$^{tm1Wjl}$/SzJ (NOD *SCID* gamma (NSG) mice provided by Charles River, www.criver.com) female mice aged 6–8 weeks for xenografts experiments. Since gender-related factors were not relevant for our study model, we used only female mice to avoid variability. Animals were maintained in sterile conditions, 12 h light/12 h dark cycle, ambient temperature 18–23 °C with 40–60% humidity. For silencing experiments, JR1 ($3.5 \times 10^6$) and RD ($5 \times 10^6$) cells infected with lentiviruses expressing either shRNAs against *SKP2* or scrambled non-targeting shRNA (shSCR) were subcutaneously injected in the left (shSKP2) and right (shSCR) flank of the mice, as reported[6]. When the masses were palpable, the measure of the masses were started twice a week for 3 weeks. For pharmacological studies, mice were subcutaneously injected in the right flank with RD ($5 \times 10^6$) or JR1 ($3.5 \times 10^6$) cells and animal groups were randomly assigned for the subsequent experiment. MLN4924/Pevonedistat dissolved in DMSO was diluted in distilled deionized water containing 30% PEG300 and 5% Tween-80 (final concentration of DMSO = 5%). MLN4924 (50 mg/kg)[17] or vehicle (the same mixture without the drug) were administrated subcutaneously once a day on a 6 days/week schedule for three weeks and, then, tumor volume was measured by caliper. Maximal tumor burden of 10% of body weight and maximal tumor size of 20 mm at the largest diameter were allowed by the ethics committee for subcutaneously injected tumors and they were not exceeded.

## Statistical analysis

The two-tailed Student's *t*-test was used for comparison between two groups; one-way ANOVA was used to analyze more than two groups; two-way ANOVA was utilized to analyze data with multiple variables. Statistical significance was set at a *p* value less than 0.05. All analyses were performed with GraphPad Prism 8.4.3 (Dotmatics, San Diego, CA, USA).

## Reporting summary

Further information on research design is available in the Nature Portfolio Reporting Summary linked to this article.

## Data availability

All RNA-seq, ChIP-seq and HiChIP data generated in this study have been deposited and publicly available in the Gene Expression Omnibus dataset under the accession number GSE241283. Affymetrix profiling data in Fig. 1a are from the following datasets: GSE66533, GSE14333, GSE108474, GSE111678, GSE14827, GSE16011, GSE26673, GSE31684, GSE32676, GSE32701, GSE34620, GSE39671, GSE42743, GSE43580, GSE64019, GSE64415, GSE7553, GSE7696, GSE87371, GSE9843, GSE2658, GSE16476, GSE9891, GSE9103, GSE2109, GSE7307 and[76]. RNA-seq analysis of cancer cell lines (Supplementary Fig. 1c) and the dependency analysis (Supplementary Fig. 1d) are from the Achilles Project Dataset (DepMap 22Q2 public + score, Chronos, https://depmap.org/portal/achilles/). ChIP-seq data (Figs. 1f and 2a, Supplementary Fig. 3a and Supplementary Fig. 4a) are from GSE83728, GSE137168, GSE29611. RNA-seq shown in Supplementary Fig. 1b is from St. Jude Children's Research Hospital PeCan database (https://pecan.stjude.cloud) and that in Supplementary Fig. 4b is from GSE52529. Source data are provided with this paper.

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

## Acknowledgements

We would like to thank Maharshi Chakraborty, Richard Sallari (Axiotl Inc, Cleveland OH, USA) and Heike Wollmann (Institute of Molecular and Cell Biology (IMCB), Agency for Science, Technology and Research (A*STAR), Singapore), for technical and bioinformatics assistance. The

Myogenin (Wright WE) and MHC (Fishman DA) antibodies were obtained from the Developmental Studies Hybridoma Bank, developed under the auspices of the NICHD and maintained by The University of Iowa, Department of Biology, Iowa City, IA 52242, USA. C.P. was supported by the Italian Ministry of University and Research (MIUR) Ph.D. fellowship (Department of Science, Roma Tre University Doctorate). We are very grateful to M. Dyer and E. Stewart from the Childhood Solid Tumor Network (CSTN) at St. Jude for PDX models used in this study. M.Co. was supported by a "Fondazione Veronesi" fellowship. This project has been funded by Associazione Italiana Ricerca sul Cancro (AIRC IG #10338, #15312 and #27794), Alleanza Contro il Cancro (ACC) Italian Network-Working Group Sarcomas, and Italian Ministry of Health Fondi 5xmille (2020-2021) and Current Research to R.R.; Alleanza Contro il Cancro (ACC) Italian Network-Working Group Sarcomas and Italian Ministry of Health Ricerca Finalizzata GR-2016-02364546 to B.D.A.; Italian Ministry of Health Ricerca Finalizzata GR-2013 02359212 to C.Q.; AIRC MFGA #22889 to E.V.; AIRC IG #24696 to F.M.; AIRC 5xmille (9962) to F.L. B.E.G. is supported through the DOD's Convergent Science Virtual Cancer Center, Reign in Sarcoma, the V Foundation and Alex's Lemonade Stand Foundation. The funders had no role in study design, data collection and analysis, decision to publish or preparation of the manuscript.

## Author contributions

S.P. and M.C. performed, interpreted and supervised aspects of all the experiments, prepared the final figures and participated in the writing of the manuscript. L.D.A. and A.P. participated in wet experiments for the revision. S.A. helped in the analyses of RNA-seq and ChIP-seq of shRNA samples under the supervision of B.E.G. D.M. participated in the study of MYOD-bound enhancer region. C.C. and C.P. helped with siRNA-mediated and immunofluorescence experiments. D.P. and P.J.H. helped with shRNA-mediated animal experiments done in USA. B.D.A. and C.Q. helped with all the animal experiments done in Italy. S.P., M.C., and Z.S.W. analyzed patients' datasets. M.P. helped with drug experiments in vitro. M.Co. initiated the experimental study. A.C. performed chromosome conformation capture analysis (3 C) experiments under the supervision of E.V. M.W. performed experiments with PDX-derived cells under the supervision of B.W.S. B.E.G. performed sequencing designed and implemented pipelines and analysis tools for RNA-seq, ChIP-seq and Hi-ChIP. S.P., M.C., and P.S. analyzed RNA-seq and ChIP-seq data. S.R. produced the GFP-SKP2 vector. C.R.V. supervised the production of Cas9-expressing cells, CRISPR vectors and CRISPR knockout experiments. C.D.S. performed immunohistochemistry experiments. R.A. helped with histology and immunohistochemistry analyses. A.W. assisted with experiments and data interpretation of pharmacological animal studies. F.M., A.D.G., G.M.M., N.C., B.W.S., C.Q., E.V., S.A.G., M.Y., G.B., E.G., and M.I. assisted with the analysis of the data and contributed to the manuscript editing. J.K., J.S., P.J.H., B.D.A., L.M., and B.E.G. made substantial contribution to the design of the study. F.L. provided support for the conception, design and results' interpretation and critically revised the manuscript. R.R. designed the study, provided overall study direction, funding, supervision and wrote and revised the manuscript. All authors critically reviewed the manuscript and approved the final version.

## Competing interests

The authors declare no competing interests.

## Additional information

Silvia Pomella[1,2,22], Matteo Cassandri[1,3,22], Lucrezia D'Archivio[1], Antonella Porrazzo[1,3], Cristina Cossetti[1], Doris Phelps[4], Clara Perrone[1], Michele Pezzella[1], Antonella Cardinale[1], Marco Wachtel[5], Sara Aloisi[6,7], David Milewski[8], Marta Colletti[1], Prethish Sreenivas[4], Zoë S. Walters[9,10], Giovanni Barillari[2], Angela Di Giannatale[1], Giuseppe Maria Milano[1], Cristiano De Stefanis[11], Rita Alaggio[12], Sonia Rodriguez-Rodriguez[13], Nadia Carlesso[13], Christopher R. Vakoc[14], Enrico Velardi[1], Beat W. Schafer[5], Ernesto Guccione[15], Susanne A. Gatz[16], Ajla Wasti[17], Marielle Yohe[18], Myron Ignatius[4], Concetta Quintarelli[1,19], Janet Shipley[9], Lucio Miele[20], Javed Khan[8], Peter J. Houghton[4], Francesco Marampon[3], Berkley E. Gryder[6], Biagio De Angelis[1], Franco Locatelli[1,21] & Rossella Rota[1] ✉

[1]Department of Hematology and Oncology, Cell and Gene Therapy, Bambino Gesù Children's Hospital, IRCCS, Roma, Italy. [2]Department of Clinical Sciences and Translational Medicine, University of Rome Tor Vergata, Rome, Italy. [3]Department of Radiological Oncological and Pathological Sciences, Sapienza University of Rome, Rome, Italy. [4]Greehey Children's Cancer Research Institute (GCCRI), UT Health Science Center, San Antonio, TX, USA. [5]Department of Oncology and Children's Research Center, University Children's Hospital, Zurich, Switzerland. [6]Department of Genetics and Genome Sciences, Case Western

Reserve University, Cleveland, OH, USA. [7]Department of Pharmacy and Biotechnology, University of Bologna, Bologna, Italy. [8]Oncogenomics Section, Genetics Branch, National Cancer Institute, NIH,, Bethesda, MD, USA. [9]Sarcoma Molecular Pathology, Divisions of Molecular Pathology, The Institute of Cancer Research, London, UK. [10]School of Cancer Sciences, Faculty of Medicine, University of Southampton, Southampton, UK. [11]Histology-Core Facility, Bambino Gesu' Children's Hospital, IRCCS, Rome, Italy. [12]Department of Pathology Unit, Department of Laboratories, Bambino Gesù Children's Hospital, IRCCS, Rome, Italy. [13]Department of Stem Cell and Regenerative Medicine, City of Hope National Medical Center, Duarte, CA, USA. [14]Cold Spring Harbor Laboratory, Cold Spring Harbor, NY, USA. [15]Center for Therapeutics Discovery, Department of Oncological Sciences and Pharmacological Sciences, Tisch Cancer Institute, Icahn School of Medicine at Mount Sinai, New York, NY, USA. [16]Institute of Cancer and Genomic Sciences, University of Birmingham, Birmingham, West Midlands, UK. [17]Children and Young People's Unit, The Royal Marsden NHS Foundation Trust and Institute of Cancer Research, Sutton, UK. [18]Laboratory of Cell and Developmental Signaling, National Cancer Institute, NIH, Frederick, MD, USA. [19]Department of Clinical Medicine and Surgery, University of Naples Federico II, Naples, Italy. [20]Department of Genetics, Louisiana State University Health Sciences Center, New Orleans, LA, USA. [21]Department of Life Sciences and Public Health, Catholic University of the Sacred Heart, Rome, Italy. [22]These authors contributed equally: Silvia Pomella, Matteo Cassandri. ✉e-mail: rossella.rota@opbg.net

