## [Peer Review File · Nature Communications]

MYOD-SKP2 axis boosts tumorigenesis in fusion negative rhabdomyosarcoma by preventing differentiation through p57Kip2 targetingREVIEWER COMMENTS

Reviewer #1 (Remarks to the Author):

The authors seek to understand the molecular mechanisms underlying the biological functions of the MYOD oncoprotein to promote rhabdomyosarcoma by transcriptionally activating Skp2 E3 ligase to target p27 and p57 for ubiquitination mediated destruction. It is clearly written, and the authors have utilized various genetic and cell biological approaches to gather strong experimental evidence. However, additional in-depth investigation should be carried out to validate the detailed mechanisms, and the following concerns should be addressed.

- 1). Figure 1e: it will be nice for the authors to include IB MYOD.
- 2). Figure 1c, 1e, it will be nice to include more than two independent siMYOD to avoid off-target issues.
- 3). Figure 3a: it will be nice to include more than two independent siSKP2, and will be nice to include other known Skp2 substrates such as FOXO1.
- 4). Figure 3g: it will be nice for the authors to include IB p21 and IB p57.
- 5). Figure 4a: will ectopic expression of p57 phenocopy Skp2 depletion? Have the authors also examined the effects of Skp2 inhibitors?
- 6). Figure 4f: as Skp2 depletion did not affect MyoD mRNA but dramatically elevate MYOD protein levels, have the authors examined the possibility of MYOD being a ubiquitin substrate of Skp2? Does Skp2 binds MYOD and promote MYOD ubiquitination and if this process depends on pSer200-MYOD event?
- 7). Figure 6d: it will be nice for the authors to examine if Skp2 inhibitors can also reduce cellular transformation abilities of the cell lines they tested.
- 8). Figure 7a: it will be nice for the authors to include IB MYOD.
- 9). Other transcriptional factors such as E2F1 have been reported to activate Skp2, will E2F1 cooperates with MYOD to promote Skp2 transcription in rhabdomyosarcoma, or MYOD is the major driver of Skp2 overexpression in rhabdomyosarcoma?

Reviewer #2 (Remarks to the Author):

Review to the manuscript:

“A MYOD-SKP2 axis boosts tumorigenesis in fusion negative rhabdomyosarcoma by preventing differentiation through p57Kip2 targeting”
By Pomella et al.

In this elegant and thorough study Pomella et al reveal a novel regulatory circuit that is imperative for the progression and survival of Fusion-negative rhabdomyosarcoma (FN-RMS), the most common soft tissue sarcoma in pediatric patients. These cancerous cells overexpress the oncogenic E3-ubiquitin ligase, SKP2, at the highest levels among a diverse group of pediatric and adult malignancies. The accelerated induction of the gene is demonstrated to be driven by MYOD binding to the intronic enhancer of SKP2, and the expressed enzyme directly targets P27kip1 and p57kip2 for degradation leading to

progression of cell cycle and stemness maintenance. By employing several orthogonal approaches, the authors explored the reciprocal dynamics and established that depletion of SKP2 expedites accumulation of p21Cip1, p27Kip1, and p57Kip2, resulting in cell cycle arrest, concomitantly with MYOD stabilization that induces Myogenin-dependent differentiation. Interestingly, Pevonedistat, an indirect inhibitor of SKP2, was shown to lower MYOG levels required for myogenic differentiation, yet it recapitulated some of the outcomes of SKP2 depletion, and triggered tumor growth suppression and apoptotic cell death.

This study nicely exemplifies the valuable potential of basic research combined with translational studies. The presented study is innovative and significant to the field of pediatric oncology. It well communicates with the established RMS literature and logically adheres to the already established findings of previous reports. The study is robust, utilizes multiple orthogonal scientific approaches, and was performed on a large cohort of tumor cell lines yielding overall consistent results that put together into a coherent story. The work supports the authors' conclusions and rational. I did not find major experimental flaws that necessitate further experimental work. In addition, data analysis seemed to be adequate and suitable for the authors' assertions and summary.

I did however spot a few issues related to the sometimes-cumbersome writing style, and have a few suggestions that may better the figure presentation.

I recommend publishing this manuscript upon fixing the minor issues I raised.

1. The following sentence needs to be altered as the discussed concept is only hypothetical (suggestive) and wasn't tested in practice, thus should be slightly tuned down. In addition, the grammar is a bit problematic:

"This suggests the presence of a core regulatory (CR) Transcription Factors (TF)s complex on the SKP2 locus in FP-RMS containing both PAX3-FOXO1 and MYOD explaining the more elevated expression of SKP2 in FP-RMS compared to FN-RMS, and was in line with data showing that PAX3-FOXO1 regulates SKP2 expression in FP-RMS".

A suggestive moderate version may be:

"This suggests the presence of a core regulatory (CR) Transcription Factors (TF)s complex on the SKP2 locus in FP-RMS containing both PAX3-FOXO1 and MYOD could propel the more elevated expression of SKP2 in FP-RMS compared to FN-RMS, in agreement with data showing that PAX3-FOXO1 regulates SKP2 expression in FP-RMS".

2. The following sentence is too long and should be separated into two, in addition, the word "on" should be altered to "of":

"Consistently, the analysis of publicly available RNA-seq data on human myoblasts induced to differentiate in vitro showed that the transcript levels of MYOD1 and SKP2 increase within the first 24 hours (24h) and then decrease in parallel, returning to their starting levels and even lower in later phases of differentiation, unlike those of MYOG, the master MYOD gene target and crucial inducer of differentiation (Fig. S2e)."

A suggestive version:

"Consistently, the analysis of publicly available RNA-seq data of human myoblasts induced to differentiate in vitro showed that the transcript levels of MYOD1 and SKP2 increase within the first 24 hours (24h) and then concomitantly decrease, returning to their starting levels

and further lower during later phases of differentiation. This, in contrast to MYOG, the chief gene target of MYOD and a crucial inducer of myogenic differentiation, whose expression levels remain elevated also by day 3 of differentiation (Fig. S2e)”.

3. The following sentence also needs to be split into two:

“Moreover, as shown in Figs. S2f-h, retrovirus-mediated expression of exogenous MyoD in murine C3H/10T1/2 fibroblasts, a well described model of myogenic-like differentiation, resulted in increased SKP2 protein and mRNA levels 24h post-infection in growth/proliferation medium (GM, supplemented with 10% serum) that was maintained 24h after the shift to differentiation medium (DM, serum-free medium), returning to steady state levels after an additional 24h in DM, when the cells were fused into multi-nucleated structures that resembled muscle fibers.”

A suggestive version:

“Moreover, as shown in Figs. S2f-h, retrovirus-mediated expression of exogenous MyoD in murine C3H/10T1/2 fibroblasts, a well described model of myogenic-like differentiation, resulted in increased mRNA and protein expression of SKP2 24h post-infection in growth/proliferation medium (GM, supplemented with 10% serum) that was maintained 24h upon shifting to differentiation medium (DM, serum-free medium). SKP2 expression levels revert to lower steady state levels by 48h in DM, however, when the cells were fused into multi-nucleated structures that resembled muscle fibers.”

4. Figure S1d: The anti-correlative nature of CERES score (lower value associated with increased sensitivity, whereas higher values associated with decreased sensitivity) makes the figure a bit confusing. For clarity, it is recommended that the authors will add to the top of the graph the following designation:

At the top left side please add:

↓ Increased sensitivity to SKP2

At the top right side add:

Reduced sensitivity to SKP2 ◇

5. Figure 1f & Figure 2a:

In both figures, please add to the gene’s schematics an arrow designating the TSS location and direction of SKP2’s transcription (similarly to the manner in which it is presented in Figure 8a).

6. The following sentence should be re-written:

“To this end, we depleted SKP2 expression for 48h in the high-risk FN-RMS cell lines RD and JR1, derived from recurrent and metastatic tumor samples respectively, both with mutated p53”.

A suggestive version:

“To this end, we depleted SKP2 expression for 48h in the high-risk FN-RMS cell lines RD and JR1 (both with mutated p53), derived from recurrent and metastatic tumor samples, respectively”.

7. Figure 2h(right):

The color of the graph’s key for CH2O 1% is wrong. The current color is whitish, whereas the color of the respective bars is bluish. Please fix the key’s color to become blue.

8. Please alter the following sentence:

“The defective ability to differentiate of FN-RMS cells highly contributes to tumorigenesis”.

A suggestive version:

“The defective ability of FN-RMS to differentiate cells highly contributes to tumorigenesis”

9. Please amend the following sentence:

“The percentage of MyHC positive cells increased by approximately 7-fold, 11-fold, 21-fold and 13-fold in RD, JR1, RD18 and RH36 SKP2 siRNA cells compared to scrambled siRNA cells”

A suggestive version:

“A comparison of SKP2 siRNA treated cell to their scrambled siRNA treated counterparts indicated that percentage of MyHC positive cells increased by approximately 7-fold, 11-fold, 21-fold and 13-fold in RD, JR1, RD18 and RH36, respectively”.

10. Please alter the following sentences accordingly:

“Similar effects were observed in RD and JR1 cells after SKP2 silencing using two lentiviral vectors expressing individual short hairpin (sh)RNAs against SKP2 (shSKP2.1 and shSKP2.2), which also increased p21Cip1 and p27Kip1 expression, compared to non-targeting control shRNA (shSCR) (Figs. 4f,g). The increase of MyHC positive cells was around 12-fold and 17-fold for shSKP2.1 and 12-fold and 14-fold for shSKP2.2 in RD and JR1 cells vs shSCR cells (Fig. 4h)”.

A suggestive version:

“Similar outcomes were observed in RD and JR1 cells by employing an orthogonal method for silencing SKP2 using two lentiviral vectors expressing individual short hairpin (sh)RNAs against SKP2 (shSKP2.1 and shSKP2.2). Here as well, SKP2 shutdown promoted increased expression of p21Cip1 and p27Kip1, compared to non-targeting control shRNA (shSCR) (Figs. 4f,g), and expanded the ratio of MyHC positive cells in both, RD and JR1 cells by at least 12-fold for both shRNAs (Fig. 4h)”.

11. Figs. S5g,h. Please refer to text:

A significant enrichment in myogenesis and muscle contraction pathways was noticed among the [HOW MANY UP-REGULATED GENES?] genes up-regulated 48h after SKP2 siRNA transduction, with MYOG showing the highest induction (Figs. S5g,h).

Please designate in text how many up-regulated genes (> 1.5 log₂ fold change) were found in total and provide the entire gene list of Up-regulated gene as a supplementary table.

Reviewer #3 (Remarks to the Author):

In this study, the authors observed SKP2 is overexpressed in RMS at the highest levels among several cancers and hypothesized its expression is maintained by MYOD1. The authors showed SKP2 directly targets p27Kip1 and p57Kip2 promoting their degradation in RMS cells and SKP2 knockdown causes cell cycle arrest by enhancing p27Kip1 and promotes differentiation by increasing p57Kip2, which in turn stabilizes MYOD resulting in muscle differentiation and cell fusion. The authors further suggested that investigational NEDDylation inhibitor MLN4924 hampers SKP2 functions restraining fusion-negative RMS cell survival and tumor growth. This study suggested a MYOD-SKP2 axis crucial for the

crosstalk between transcriptional and post-translational mechanisms that contribute to RMS tumorigenesis and broaden the understanding of MYOD function. However, there are major issues needed to be addressed in the present manuscript. Especially, this study is designed based on the hypothesis that MYOD1 is an oncogene in RMS, which is unacceptable without any reasonable data.

(Major points)

1. The authors suggest the oncogenic function of MYOD1 to transcriptionally regulate the expression of SKP2. It is known that MYOD1 L122R mutation blocks wild-type MYOD1 function and bind to MYC consensus sequences acting as an oncogene to inhibit differentiation and promote proliferation. However, wild-type MYOD1 involvement in tumorigenesis is not clear. This hypothesis should be carefully confirmed. Does knock down of MYOD1 by siRNA inhibit the tumor growth and increased myogenic differentiation? Does MYOD1 overexpression, vice versa, increase the tumor growth?

2. The authors show ChIP-seq data to suggest MYOD1 involvement in SKP2 transcription. However, it doesn't reveal how strongly MYOD1 regulates SKP2. Although si MYOD1 decrease SKP2 expression, it may be caused indirectly as a result of feedback to the dedifferentiation caused by MYOD1 depletion. Reporter gene assay may be required to convince the existence of MYOD1-SKP2 axis.

3. It seems that SKP2 is a cell cycle regulator in general irrespective of MYOD1. Therefore, inhibiting SKP2 cause growth arrest not only sarcoma cell but also normal cells suggesting that SKP2 is not a reasonable target gene. It is needed to demonstrate the growth arrest by SKP2 knockdown is correlate with the MYOD1 expression.

4. The mode of action of MLN4924 is not relevant to MYOD1 function. Is MYOD1 expression and the drug sensitivity correlate?

Reviewer #4 (Remarks to the Author):

Review of Pomella et al.

The authors here have identified SKP2 as a critical driver of tumorigenesis in fusion negative rhabdomyosarcoma (FN-RMS) and acting downstream of MYOD. The authors have utilized publicly available genome-wide datasets both from their previous work and from others to establish the transcriptional regulation of SKP2 through MYOD. The study design and experiments are good and done in a systematic manner to show the MYOD-SKP2 axis that contributes to the tumorigenic phenotype in FN-RMS. Overall, the work performed here is satisfactory and novel in dissecting the MYOD-SKP2 signaling axis in FN-RMS. I have few concerns that should be addressed to make the findings more robust and clearer from the mechanistic point of view and outreach of this research.

The major points of concern are:

1. The authors mention that SKP2 directly interacts with p27Kip1 and targets it for proteasomal degradation, and no interaction was detected for p21Cip1. Although both p21Cip1 and p27Kip1 protein levels increased post-treatment with proteasome inhibitor MG132. How do the authors explain this discrepancy?
2. SKP2 depletion in FN-RMS cells resulted in increased MYOD and and CDKI p57Kip2.

What is the consequence of MYOD binding to chromatin at the SKP2 regulatory elements such as the intronic enhancer? Can the authors perform ChIP-seq or ChIP-qPCR to show the effect of MYOD binding after SKP2 depletion? Although ChIP-qPCR could be more direct for that particular SKP2 regulatory region but the genome-wide ChIP-seq after SKP2 silencing could also pinpoint the target genes for reduced stemness and tumorigenicity in RD and JR1 cells.

3. The authors used β -gal staining assay to show induction of senescence upon SKP2 depletion. Could the authors validate the same on the molecular levels showing the levels of any senescent marker genes for example SASP protein either by quantitative real time PCR from the available cDNA or by western blot.

4. The authors have shown effect on cell proliferation upon SKP2 depletion in cell lines and the same effect is also seen in the reduced tumor volumes after SKP2 is silenced by shRNA or using the NAE inhibitor LN4924. Since the IHC sections have been made and stained for the relevant targets but it was surprising that the authors haven't stained for the proliferation marker Ki67 in these sections. The Ki67 staining for the IHC sections shown in Fig 6k and 7c would make it clear at the molecular level.

Response to Reviewers

Reviewer #1 (Remarks to the Author):

The authors seek to understand the molecular mechanisms underlying the biological functions of the MYOD oncoprotein to promote rhabdomyosarcoma by transcriptionally activating Skp2 E3 ligase to target p27 and p57 for ubiquitination mediated destruction. It is clearly written, and the authors have utilized various genetic and cell biological approaches to gather strong experimental evidence. However, additional in-depth investigation should be carried out to validate the detailed mechanisms, and the following concerns should be addressed.

We thank the Reviewer for the positive evaluation of our work and suggestions for improvement.

1). *Figure 1e: it will be nice for the authors to include IB MYOD.*

We have performed immunoblot for MYOD on the panel of RMS cell lines as requested. However, since in Figure 1, and in the corresponding paragraph, MYOD has not yet been mentioned, we included this result as Fig. S2a.

2). *Figure 1c, 1e, it will be nice to include more than two independent siMYOD to avoid off-target issues.*

We have performed MYOD silencing using 2 additional independent MYOD siRNAs in all the four RMS cell lines. The results for RD and RH4 cell lines are shown in Fig. S2b,c while those for JR1 and RH30 are included in the Ancillary Fig. 1 for the Reviewers. MYOD was down-regulated together with SKP2 at the protein and transcript levels starting from 24h post-transfection, supporting a direct regulation by MYOD. In the original version we also have used a CRISPR/Cas9 approach to knockout MYOD and showed the same effects on SKP2 expression (now reported in Fig. S2d,e). Altogether, these results suggest a lack of off-target effects.

3). *Figure 3a: it will be nice to include more than two independent siSKP2 and will be nice to include other known Skp2 substrates such as FOXO1.*

We completely agree with the Reviewer that it is mandatory to demonstrate that our approach is not the result of off-target effects. For this reason, in our initial submission, we used a validated siRNA sequence against SKP2 (by Sigma-Aldrich) in Fig. 3a and we also validated the results using a siRNA pool containing 4 different individual SKP2 siRNA sequences (by Dharmacon) (Original Fig. S3a and now Fig. S4a).

In addition, we infected our cells with two lentiviral vectors each expressing an individual SKP2 shRNA sequence (by Sigma-Aldrich) different from all the others used as siRNAs (Fig. 4f). In all these cases, we detected SKP2 down-regulation associated with both p21 and p27 protein levels increase. Therefore, we are confident our results are not due to off-target effects. We have revised the main text in the Results section to better clarify this point.

As suggested by the Reviewer, we did immunoblot for FOXO1 and added it to Fig. 3a. We found increased levels of the protein after SKP2 silencing compared to scrambled siRNA cells validating the SKP2 down-regulation effects.

4). *Figure 3g: it will be nice for the authors to include IB p21 and IB p57.*

We have performed immunoblot for p21 and included the results in Fig. 3g. The results show that p27 knockdown reduced p21 increase due to SKP2 silencing. In addition, it also slightly decreased p21 levels in basal conditions compared to scrambled siRNA.

These results suggest that p27 could indirectly regulate p21 expression in FN-RMS with a mechanism that deserves further investigations.

Moreover, we also investigated the expression levels of p57 by immunoblot. We have included the results in the Ancillary Fig. 2 for the Reviewers since, at this point of the manuscript, we have not yet presented data on p57 and it would be difficult to explain why this detection was made. In fact, the study of p57 expression was done related to its role in cells differentiation starting from Fig. 4. Moreover, no significant effects of p27 depletion can be seen on p57 levels.

5). *Figure 4a: will ectopic expression of p57 phenocopy Skp2 depletion?*

To address the point referring to Fig. 4, we have overexpressed an exogenous p57/*CDKN1C* gene in RD and JR1 FN-RMS cell lines and evaluated whether higher p57 levels would be sufficient to induce differentiation of FN-RMS cell lines mirroring SKP2 down-regulation. The results have been reported in Fig. S6. Notably, the protein and mRNA levels of MYOD and MYOG increased in p57-vector compared to empty vector cells. In agreement, the p57-overexpressing cells showed a tendency to differentiate compared to empty vector cells, albeit more mildly than observed after SKP2 depletion. These results suggest that p57 positively modulates differentiation in FN-RMS cells and this function is even more evident in a molecular setting where the brakes for differentiation have been relieved, such as when SKP2 is depleted. These findings have been reported and discussed in the revised manuscript.

Have the authors also examined the effects of Skp2 inhibitors?

As an indirect SKP2 inhibitor with translational potentiality we used MLN4924/Pevonedistat, which is being investigated in several clinical trials of different tumor types (see Results section and Figs. 7 and S10). However, we have performed new experiments using the *in vitro* validated SKP2 inhibitor SMIP004 (Ref. 28 Revised Manuscript).

SMIP004 treatment decreased SKP2 protein levels and increased p21 and p27 protein and mRNA levels in RD and JR1 cells 48h post-treatment (Fig. 3i-n). This was associated with slowdown of cell proliferation and G1 cell cycle arrest compared to vehicle. Moreover, it promoted MYOD, MYOG and p57 protein and mRNA levels enhancement and the expression of the late differentiation marker MyHC (Fig. 4i-l).

Altogether, these results show that pharmacological inhibition of SKP2 with SMIP004 mirrors the anti-proliferative and pro-differentiation effects of genetic depletion.

In parallel, since in the original submission we investigated the *in vitro* effects of MLN4924 on cell proliferation, we performed new experiments to assess cell cycle distribution of the RD and JR1 cell lines also under MLN4924. The results in Ancillary Fig. 3 for Reviewers show that, conversely to SKP2 silencing and SMIP004, treatment with MLN4924 resulted in the accumulation of cells in G2 phase as compared to vehicle. This is in agreement with the evidence that the drug leads to down-regulation of MYOG and apoptosis rather than differentiation, as reported and discussed in the original version of the manuscript.

6). *Figure 4f: as Skp2 depletion did not affect MyoD mRNA but dramatically elevate MYOD protein levels, have the authors examined the possibility of MYOD being a ubiquitin substrate of Skp2? Does Skp2 binds MYOD and promote MYOD ubiquitination and if this process depends on pSer200-MYOD event?*

As requested, we have performed Co-IP for SKP2 and MYOD in both RD and JR1 FN-RMS cells and found no interaction between the two proteins (Ancillary Fig. 4 for Reviewers). These experiments demonstrate that MYOD is not a SKP2 substrate in FN-RMS cells.

This is in line with data showing that MyoD does not interact with Skp2 in murine myoblasts (Reference for Rebuttal ¹).

Of note, in the original and revised manuscript we showed that, in addition to the increase in MYOD protein levels, SKP2 depletion resulted in the rise of *MYOD1* mRNA levels at least in RD and JR1 cells (now Figs. 4a,b).

Thus, it is conceivable that the enhancement of MYOD levels after SKP2 depletion could be the result of, at least, two concurrent phenomena: a post-transcriptional protein stabilization by p57 and an increment of transcriptional auto-induction by MYOD its-self (Ref. n. 45). However, since Myod is regulated by the proteasome in myoblasts, we cannot exclude that SKP2 knockdown could indirectly or directly modulate other E3 ubiquitin ligases that, in turn, participate to the post-transcriptional regulation of MYOD levels ¹.

7). *Figure 6d: it will be nice for the authors to examine if Skp2 inhibitors can also reduce cellular transformation abilities of the cell lines they tested.*

To evaluate this aspect, we have performed new experiments treating RD and JR1 cells with the two SKP2i, SMIP004 and MLN4924. The results are reported in Fig. S9a-d and Fig. S10d-g, respectively. The ability to grow as colonies in an anchorage-independent assay and to form rhabdospheres when cultured in a serum-free stem cell medium were both compromised by the two agents compared to vehicle treatment, mirroring the effects of SKP2 genetic depletion.

8). *Figure 7a: it will be nice for the authors to include IB MYOD.*

The protein levels of MYOD after treatment with MLN4924 for both RD and JR1 FN-RMS cell lines were already reported in our original version in Fig. S7b (now Fig. 7c).

9). *Other transcriptional factors such as E2F1 have been reported to activate Skp2, will E2F1 cooperates with MYOD to promote Skp2 transcription in rhabdomyosarcoma, or MYOD is the major driver of Skp2 overexpression in rhabdomyosarcoma?*

To clarify this point, we have performed *E2F1* silencing in RD and JR1 FN-RMS as well as in RH4 and RH30 P3F-RMS cell lines using two individual validated siRNAs (Ancillary Fig. 5 for Reviewers). The two independent experiments show no transcript and/or protein levels modulation of SKP2 after *E2F1* silencing in our tumor cell context. However, we agree that *SKP2* expression has been shown to be modulated by several TFs in different cell contexts. In addition to *E2F1*, other TFs are GA-binding protein (GABP) and NUCKS1, which induce *SKP2* expression ^{2,3}, and STAT1 that, conversely, represses the gene ⁴.

Thus, we cannot rule out that other TFs relevant to the FN-RMS context could participate with MYOD in the transcriptional regulation of *SKP2*. However, being MYOD a lineage-specific TF highly expressed in FN-RMS, also based on our data it could be considered as one of the major drivers.

Reviewer #2 (Remarks to the Author):

Review to the manuscript:

“A MYOD-SKP2 axis boosts tumorigenesis in fusion negative rhabdomyosarcoma by preventing differentiation through p57Kip2 targeting”

By Pomella et al.

In this elegant and thorough study Pomella et al reveal a novel regulatory circuit that is imperative for the progression and survival of Fusion-negative rhabdomyosarcoma (FN-RMS), the most common soft tissue sarcoma in pediatric patients. These cancerous cells overexpress the oncogenic E3-ubiquitin ligase, SKP2, at the highest levels among a diverse group of pediatric and adult malignancies. The accelerated induction of the gene is demonstrated to be driven by MYOD binding to the intronic enhancer of SKP2, and the expressed enzyme directly targets P27kip1 and p57kip2 for degradation leading to progression of cell cycle and stemness maintenance. By employing several orthogonal approaches, the authors explored the reciprocal dynamics and

established that depletion of SKP2 expedites accumulation of p21Cip1, p27Kip1, and p57Kip2, resulting in cell cycle arrest, concomitantly with MYOD stabilization that induces Myogenin-dependent differentiation. Interestingly, Pevonedistat, an indirect inhibitor of SKP2, was shown to lower MYOG levels required for myogenic differentiation, yet it recapitulated some of the outcomes of SKP2 depletion, and triggered tumor growth suppression and apoptotic cell death.

This study nicely exemplifies the valuable potential of basic research combined with translational studies. The presented study is innovative and significant to the field of pediatric oncology. It well communicates with the established RMS literature and logically adheres to the already established findings of previous reports. The study is robust, utilizes multiple orthogonal scientific approaches, and was performed on a large cohort of tumor cell lines yielding overall consistent results that put together into a coherent story. The work supports the authors' conclusions and rational. I did not find major experimental flaws that necessitate further experimental work. In addition, data analysis seemed to be adequate and suitable for the authors' assertions and summary.

I did however spot a few issues related to the sometimes-cumbersome writing style, and have a few suggestions that may better the figure presentation.

I recommend publishing this manuscript upon fixing the minor issues I raised.

We thank the Reviewer for the positive evaluation of the manuscript and for the suggestions to improve the writing which have been very valuable.

1. The following sentence needs to be altered as the discussed concept is only hypothetical (suggestive) and wasn't tested in practice, thus should be slightly tuned down. In addition, the grammar is a bit problematic:

"This suggests the presence of a core regulatory (CR) Transcription Factors (TF)s complex on the SKP2 locus in FP-RMS containing both PAX3-FOXO1 and MYOD explaining the more elevated expression of SKP2 in FP-RMS compared to FN-RMS, and was in line with data showing that PAX3-FOXO1 regulates SKP2 expression in FP-RMS".

A suggestive moderate version may be:

"This suggests the presence of a core regulatory (CR) Transcription Factors (TF)s complex on the SKP2 locus in FP-RMS containing both PAX3-FOXO1 and MYOD could propel the more elevated expression of SKP2 in FP-RMS compared to FN-RMS, in agreement with data showing that PAX3-FOXO1 regulates SKP2 expression in FP-RMS".

We have revised the manuscript as suggested.

2. The following sentence is too long and should be separated into two, in addition, the word "on" should be altered to "of":

"Consistently, the analysis of publicly available RNA-seq data on human myoblasts induced to differentiate in vitro showed that the transcript levels of MYOD1 and SKP2 increase within the first 24 hours (24h) and then decrease in parallel, returning to their starting levels and even lower in later phases of differentiation, unlike those of MYOG, the master MYOD gene target and crucial inducer of differentiation (Fig. S2e)."

A suggestive version:

"Consistently, the analysis of publicly available RNA-seq data of human myoblasts induced to differentiate in vitro showed that the transcript levels of MYOD1 and SKP2 increase within the first 24 hours (24h) and then concomitantly decrease, returning to their starting levels and further lower during later phases of differentiation. This, in contrast to MYOG, the chief gene target of MYOD and a crucial inducer of myogenic differentiation, whose expression levels remain elevated also by day 3 of differentiation (Fig. S2e)".

We have revised the manuscript as suggested.

3. The following sentence also needs to be split into two:

“Moreover, as shown in Figs. S2f-h, retrovirus-mediated expression of exogenous MyoD in murine C3H/10T1/2 fibroblasts, a well described model of myogenic-like differentiation, resulted in increased SKP2 protein and mRNA levels 24h post-infection in growth/proliferation medium (GM, supplemented with 10% serum) that was maintained 24h after the shift to differentiation medium (DM, serum-free medium), returning to steady state levels after an additional 24h in DM, when the cells were fused into multi-nucleated structures that resembled muscle fibers.”

A suggestive version:

“Moreover, as shown in Figs. S2f-h, retrovirus-mediated expression of exogenous MyoD in murine C3H/10T1/2 fibroblasts, a well described model of myogenic-like differentiation, resulted in increased mRNA and protein expression of SKP2 24h post-infection in growth/proliferation medium (GM, supplemented with 10% serum) that was maintained 24h upon shifting to differentiation medium (DM, serum-free medium). SKP2 expression levels revert to lower steady state levels by 48h in DM, however, when the cells were fused into multi-nucleated structures that resembled muscle fibers.”

We have revised the manuscript as suggested.

4. Figure S1d: The anti-correlative nature of CERES score (lower value associated with increased sensitivity, whereas higher values associated with decreased sensitivity) makes the figure a bit confusing. For clarity, it is recommended that the authors will add to the top of the graph the following designation:

At the top left side please add:

↙ Increased sensitivity to SKP2

At the top right side add:

↘ Reduced sensitivity to SKP2

We thank the Reviewer for his/her suggestion. We have modified the Figure accordingly.

5. Figure 1f & Figure 2a:

In both figures, please add to the gene’s schematics an arrow designating the TSS location and direction of SKP2’s transcription (similarly to the manner in which it is presented in Figure 8a).

The Figures have been modified as suggested.

6. The following sentence should be re-written:

“To this end, we depleted SKP2 expression for 48h in the high-risk FN-RMS cell lines RD and JR1, derived from recurrent and metastatic tumor samples respectively, both with mutated p53”.

A suggestive version:

“To this end, we depleted SKP2 expression for 48h in the high-risk FN-RMS cell lines RD and JR1 (both with mutated p53), derived from recurrent and metastatic tumor samples, respectively”.

We have revised the manuscript as suggested.

7. Figure 2h(right):

The color of the graph's key for CH2O 1% is wrong. The current color is whitish, whereas the color of the respective bars is bluish. Please fix the key's color to become blue.

We thank the Reviewer for noticing this error that has been, now, fixed.

8. Please alter the following sentence:

"The defective ability to differentiate of FN-RMS cells highly contributes to tumorigenesis".

A suggestive version:

"The defective ability of FN-RMS to differentiate cells highly contributes to tumorigenesis"

We have revised the manuscript as suggested.

9. Please amend the following sentence:

"The percentage of MyHC positive cells increased by approximately 7-fold, 11-fold, 21-fold and 13-fold in RD, JR1, RD18 and RH36 SKP2 siRNA cells compared to scrambled siRNA cells"

A suggestive version:

"A comparison of SKP2 siRNA treated cell to their scrambled siRNA treated counterparts indicated that percentage of MyHC positive cells increased by approximately 7-fold, 11-fold, 21-fold and 13-fold in RD, JR1, RD18 and RH36, respectively".

We have revised the manuscript as suggested.

10. Please alter the following sentences accordingly:

"Similar effects were observed in RD and JR1 cells after SKP2 silencing using two lentiviral vectors expressing individual short hairpin (sh)RNAs against SKP2 (shSKP2.1 and shSKP2.2), which also increased p21Cip1 and p27Kip1 expression, compared to non-targeting control shRNA (shSCR) (Figs. 4f,g). The increase of MyHC positive cells was around 12-fold and 17-fold for shSKP2.1 and 12-fold and 14-fold for shSKP2.2 in RD and JR1 cells vs shSCR cells (Fig. 4h)".

A suggestive version:

"Similar outcomes were observed in RD and JR1 cells by employing an orthogonal method for silencing SKP2 using two lentiviral vectors expressing individual short hairpin (sh)RNAs against SKP2 (shSKP2.1 and shSKP2.2). Here as well, SKP2 shutdown promoted increased expression of p21Cip1 and p27Kip1, compared to non-targeting control shRNA (shSCR) (Figs. 4f,g), and expanded the ratio of MyHC positive cells in both, RD and JR1 cells by at least 12-fold for both shRNAs (Fig. 4h)".

We have revised the manuscript as suggested.

11. Figs. S5g,h. Please refer to text:

A significant enrichment in myogenesis and muscle contraction pathways was noticed among the [HOW MANY UP-REGULATED GENES?] genes up-regulated 48h after SKP2 siRNA transduction, with MYOG showing the highest induction (Figs. S5g,h).

Please designate in text how many up-regulated genes (> 1.5 log₂ fold change) were found in total and provide the entire gene list of Up-regulated gene as a supplementary table.

In the original submission we presented data of RNA-seq after SKP2 siRNA in RD cells. In the revised version we have done a new RNA-seq assay after SKP2 silencing using shSKP2.2 and

shSCR in both FN-RMS cell lines RD and JR1 in order to identify commonly modulated genes. Based on these new experiments, we have revised the paragraph enclosing the results in Fig. S8 and provided a new gene list of total up-regulated (> 1.3 Log2 fold change) and down-regulated (< 0.7 Log2 fold change) genes in Supplementary Table 1.

Moreover, we have also provided the lists of differentially modulated genes (same criteria as above) for myogenesis, stemness and senescence (Fig. S8) in Supplementary Table 2.

Reviewer #3 (Remarks to the Author):

In this study, the authors observed SKP2 is overexpressed in RMS at the highest levels among several cancers and hypothesized its expression is maintained by MYOD1. The authors showed SKP2 directly targets p27Kip1 and p57Kip2 promoting their degradation in RMS cells and SKP2 knockdown causes cell cycle arrest by enhancing p27Kip1 and promotes differentiation by increasing p57Kip2, which in turn stabilizes MYOD resulting in muscle differentiation and cell fusion. The authors further suggested that investigational NEDDylation inhibitor MLN4924 hampers SKP2 functions restraining fusion-negative RMS cell survival and tumor growth. This study suggested a MYOD-SKP2 axis crucial for the crosstalk between transcriptional and post-translational mechanisms that contribute to RMS tumorigenesis and broaden the understanding of MYOD function. However, there are major issues needed to be addressed in the present manuscript. Especially, this study is designed based on the hypothesis that MYOD1 is an oncogene in RMS, which is unacceptable without any reasonable data.

We thank the Reviewer very much for allowing us to clarify this crucial point. We absolutely agree with the Reviewer that MYOD1 cannot be considered a “primary” oncogene since it is not a “driver” of RMS.

However, *MYOD1* is among the most RMS-selective gene dependency, as demonstrated analyzing a genome-wide CRISPR/Cas9 somatic knockout (KO) screen on the survival of tumor cell lines: MYOD1 is crucial in supporting RMS cell survival (Achilles project, <https://depmap.org/portal/achilles>) (Ancillary Fig. 6 for Reviewers).

This data has also been reported in a recent study of the group of Kimberly Stegmaier investigating “genetic vulnerabilities unique of a specific cancer type” in pediatric cancers (⁵ Fig. 1a, right panel). Pediatric cancers have a low mutational burden and it is now recognized that, in addition to the driver oncogenes on which the tumor cells depend (oncogene addiction), tumor survival mechanisms are often pre-programmed in the cells of origin. Specifically, these deregulated epigenetic mechanisms hijack wild-type lineage-specific Transcription Factors (TFs) into tumor-specific core regulatory circuits (CRCs) needed for tumorigenesis, resulting in a “lineage dependency” (reviewed in ^{6,7}).

Therefore, we apologize for having utilized the inappropriate term “oncogenic” in two sentences about MYOD. We used the term “pro-tumorigenic” regarding MYOD **to indicate a TF that reinforces an oncogenic driver program** clarifying this point throughout the revised manuscript.

In addition, we would like to point out that our work was specifically focused on the study of the role of SKP2 in FN-RMS, which remained unknown until now, by dissecting the molecular pathways implicated.

In this scenario, we unexpectedly discovered that MYOD was involved in SKP2 functions as both an upstream and downstream molecular player. Our data demonstrating that MYOD regulates the expression of an oncogenic E3 ubiquitin ligase like SKP2 add a piece of evidence on the intricate molecular networks sustaining RMS tumorigenesis.

(Major points)

1. *The authors suggest the oncogenic function of MYOD1 to transcriptionally regulate the expression of SKP2. It is known that MYOD1 L122R mutation blocks wild-type MYOD1 function and bind to MYC consensus sequences acting as an oncogene to inhibit differentiation and promote proliferation. However, wild-type MYOD1 involvement in tumorigenesis is not clear. This hypothesis should be carefully confirmed. Does knock down of MYOD1 by siRNA inhibit the tumor growth and increased myogenic differentiation?*

This aspect has already been investigated by David Langenau's group showing that MYOD1 is needed for "sustained tumor growth" of RD and SMS-CTR FN-RMS cells *in vitro* and *in vivo* since its depletion results in tumor cell death (Ref. 7 Revised Manuscript).

However, as requested by the Reviewer we have performed new functional experiments by silencing MYOD1 in RD and JR1 FN-RMS cell lines. In line with the dependency of RMS cells on this TF (see the response to comment above), MYOD-depleted cells showed reduced proliferation and accumulated in the G1 cell cycle phase reducing S phase compared to scrambled siRNA cells (Ancillary Fig. 7a,b for Reviewers).

These results confirm those obtained by the group of Langenau about MYOD depletion (Ref. 7 Revised Manuscript) as well as the data on MYOD knockout in RMS cell lines of the DepMap portal (as reported in the previous comment to the same Reviewer).

Both mRNA and protein levels of SKP2 decreased in MYOD siRNA cells, as already shown in Figs. 2 and S2 and in Ancillary Fig. 1 for Reviewers. P21 and p27 protein levels increased in MYOD1-silenced cells independently of their mRNA levels, which is in agreement with the down-regulation of SKP2 (Ancillary Fig. 7c,d for Reviewers).

However, MYOD-silenced FN-RMS cells displayed downregulation of the MYOD chief direct target MYOG, one of the master muscle regulatory factors and, thus, are unable to differentiate (Ancillary Fig. 7e for Reviewers) (Ref. 7 Revised Manuscript; reviewed in ⁸).

Overall, our results confirm data from other labs (see above) showing that MYOD is needed for continued FN-RMS cells growth/survival, which is also in agreement with the function of SKP2.

Does MYOD1 overexpression, vice versa, increase the tumor growth?

To answer to this question, we performed MYOD overexpression in both RD and JR1 FN-RMS cell lines using an inducible viral vector expressing an exogenous MYOD1 (Ancillary Fig. 8 for Reviewers). MYOD levels were markedly increased after induction with doxycycline compared to control cells. In parallel, SKP2 was modestly augmented maybe due to the already high basal levels. No increase in cell proliferation was evident in MYOD-overexpressing cells. This result seems to be in line with a pro-survival rather than pro-proliferative role of MYOD in FN-RMS and/or could also be related to the high basal expression of the TF suggesting that a further increase is not sufficient to enhance cell proliferation.

2. *The authors show ChIP-seq data to suggest MYOD1 involvement in SKP2 transcription. However, it doesn't reveal how strongly MYOD1 regulates SKP2. Although si MYOD1 decrease SKP2 expression, it may be caused indirectly as a result of feedback to the dedifferentiation caused by MYOD1 depletion. Reporter gene assay may be required to convince the existence of MYOD1-SKP2 axis.*

We thank the Reviewer for his/her suggestion that allow us to clarify our data and strengthen the manuscript. To this end, a reporter gene assay has been performed in HEK 293T cells, not expressing MYOD1, co-transfecting either (i) a plasmid vector expressing human MYOD or an empty vector and (ii) a pGL3-promoter vector expressing a luciferase gene under the control of the identified MYOD-bound SKP2 intronic enhancer region or an empty pGL3-promoter vector (Fig. S2g). Results show luciferase induction only in cells co-transfected with MYOD-expressing vector

and pGL3 plasmid harboring the MYOD-bound SKP2 intronic enhancer suggesting MYOD-transactivation activity.

Moreover, we applied an additional approach interrogating chromatin looping mechanisms in the RD FN-RMS cell line using HiChIP technique in order to identify chromatin contacts around the SKP2 promoter at 1kb resolution. RH4 FP-RMS cell line was *de novo* assayed as well since the sequencing used for the same cell line in the original version of the manuscript was at 5 kb resolution. As shown in Fig. S2f, the data analysis confirmed an interaction between the identified MYOD-bound intronic region and the SKP2 promoter in both cell lines.

These data, together with those obtained with the chromosome conformation capture assay (3C) in the original version of the manuscript, which showed that the MYOD-bound intronic enhancer was able to bind SKP2 promoter (Fig. 2g,h), support a direct transcriptional regulation of SKP2 expression by MYOD.

3. It seems that SKP2 is a cell cycle regulator in general irrespective of MYOD1. Therefore, inhibiting SKP2 cause growth arrest not only sarcoma cell but also normal cells suggesting that SKP2 is not a reasonable target gene. It is needed to demonstrate the growth arrest by SKP2 knockdown is correlate with the MYOD1 expression.

To answer to this concern, we firstly investigated whether FN-RMS cells had increased vulnerability to SKP2 inhibition compared to a panel of normal cells. To this end, we performed new experiments: human normal myoblasts, which are the normal lineage counterpart of RMS cells and express MYOD, lung and dermal fibroblasts, which do not express MYOD, were treated in dose response experiments with the two SKP2i used in the study, SMIP004 and MLN4924, and their survival ability assessed.

Results reported in Figs. S4k and S10b show that all normal cells are considerably less sensitive to high doses of both treatments compared to FN-RMS cells suggesting a tumor-related vulnerability (see also the revised Results section).

In addition, the response to SKP2 inhibition clearly appeared not correlated to MYOD since myoblasts and fibroblasts showed a similar response.

Our data are concordant with those of the group of Singer showing that SKP2 depletion has negligible effects on normal pre-adipocyte stem cells vs myxofibrosarcoma cells (Ref. 17 Revised Manuscript). Calandrini et al. also demonstrated selective sensitivity of Malignant Rhabdoid Tumors organoids to MLN4924 with respect to healthy kidney and liver tissue organoids (Ref. 64 Revised Manuscript). They, also, reported that only the healthy small intestine organoids were affected by MLN4924 possibly due to high rate of proliferation *in vitro* but they also highlighted that no side effects related to the intestines have been identified in the first clinical studies on MLN4924.

However, we agree with the Reviewer that, under a translational point of view, it is very important to identify those SKP2 targeting compounds that can be more effective in that specific tumor context and to consider the pathological mechanisms of the targeted tumor type (Refs. 53, 67 Revised Manuscript; and reviewed in Ref. 68 Revised Manuscript). In this view, preclinical studies on selected tumors could be of help.

We, therefore, have added a comment on these aspects in the revised Discussion.

In addition, to answer to the reviewer's comment more exhaustively, we decided to elucidate the interconnection between MYOD and SKP2 in FN-RMS by silencing the two factors individually or together.

The Ancillary Fig. 9a,b for Reviewers, which partly refers to data on MYOD-silenced cells (see the response to point 1 of the same Reviewer and Ancillary Fig. 7 for Reviewers), shows that silencing

of MYOD alone vs siSCR reduced cell numbers and promoted G1 phase increment and S phase diminution more markedly than SKP2 siRNA alone in RD and JR1 cells.

This finding is in line with the crucial pro-survival role of MYOD in FN-RMS cells (see also the response to point 1 raised by the same Reviewer). SKP2 transcriptional and protein down-regulation following MYOD silencing was confirmed in Ancillary Fig. 9c,d for Reviewers.

MYOD and SKP2 co-depleted cells showed significant decrease of cell proliferation paralleled by enhancement of G1 phase and reduction of S phase cells accumulation compared to SKP2 silenced cells, while the effects were not significant vs MYOD-siRNA alone.

These findings clearly demonstrate that the effects of SKP2 on cell cycle are independent on MYOD since MYOD knockdown in a SKP2-depleted context does not reduce, but in contrast enhances, cell cycle arrest.

MYOD siRNA alone lowered MYOG protein and transcript levels, as already shown, and overcomes MYOG induction when transfected in combination with SKP2 siRNA Ancillary Fig. 9c,d for Reviewers. Consequently, reducing MYOD levels not only does not induce differentiation but also counteracts SKP2 depletion-dependent cell differentiation (Ancillary Fig. 9e,d for Reviewers). These findings clearly demonstrate that MYOD appears to be necessary for the pro-differentiation effects of SKP2 inhibition.

This is in line with the evidence that, although p57 levels were higher in MYOD/SKP2 co-depleted cells, they were not sufficient to support differentiation in the absence (or reduction) of MYOD.

Moreover, being the percentage of MYOD depleted cells arrested in the G1 phase of the cell cycle significantly higher compared to that of SKP2 siRNA cells, our results also suggest that MYOD regulates FN-RMS cell survival also modulating other pathways in addition to SKP2, as we have recently demonstrated (Ref. 6 Revised Manuscript).

In conclusion, our new results validate SKP2 as a tumor survival vulnerability in FN-RMS.

4. The mode of action of MLN4924 is not relevant to MYOD1 function. Is MYOD1 expression and the drug sensitivity correlate?

Although this point has been clarified in the above response to point 3, we performed Pearson correlation analysis taking advantage of the DepMap portal (<https://depmap.org/portal/>) using the 23Q2 public dataset for gene expression and Drug Sensitivity Area Under the Curve (AUC) dataset for drug sensitivity data, which includes 721 cancer lines (CTD2, Broad Institute) (Ancillary Figures 10a,b for Reviewers). The analysis revealed no significant correlation between *MYOD1* expression and the response to MLN4924 while a significant correlation between SKP2 expression and the drug AUC was observed among all the cancer cell lines analyzed.

Reviewer #4 (Remarks to the Author):

Review of Pomella et al.

The authors here have identified SKP2 as a critical driver of tumorigenesis in fusion negative rhabdomyosarcoma (FN-RMS) and acting downstream of MYOD. The authors have utilized publicly available genome-wide datasets both from their previous work and from others to establish the transcriptional regulation of SKP2 through MYOD. The study design and experiments are good and done in a systematic manner to show the MYOD-SKP2 axis that contributes to the tumorigenic phenotype in FN-RMS. Overall, the work performed here is satisfactory and novel in dissecting the MYOD-SKP2 signaling axis in FN-RMS. I have few concerns that should be addressed to make the findings more robust and clearer from the mechanistic point of view and outreach of this research.

We thank the Reviewer for the positive overall evaluation of the manuscript and the suggestions for improvement.

The major points of concern are:

1. The authors mention that SKP2 directly interacts with p27Kip1 and targets it for proteasomal degradation, and no interaction was detected for p21Cip1. Although both p21Cip1 and p27Kip1 protein levels increased post-treatment with proteasome inhibitor MG132. How do the authors explain this discrepancy?

As reported in our original submission, we were unable to detect an interaction between SKP2 and p21. However, we cannot exclude that the basal levels of p21 bound to SKP2 are under the threshold of antibody detection in our cell context.

It could be also possible that the enhancement of the p21 protein levels before the increase of its transcripts levels was due to protein stabilization through the modulation of other factors. P21, indeed, is subjected to context-dependent intense post-transcriptional modulation by a number of E3 ligases⁹⁻¹².

2. SKP2 depletion in FN-RMS cells resulted in increased MYOD and and CDKI p57Kip2. What is the consequence of MYOD binding to chromatin at the SKP2 regulatory elements such as the intronic enhancer? Can the authors perform ChIP-seq or ChIP-qPCR to show the effect of MYOD binding after SKP2 depletion? Although ChIP-qPCR could be more direct for that particular SKP2 regulatory region but the genome-wide ChIP-seq after SKP2 silencing could also pinpoint the target genes for reduced stemness and tumorigenicity in RD and JR1 cells.

As requested by the Reviewer, we performed ChIP-seq for MYOD and the H3K27ac enhancer mark in shSKP2.2 and shSCR RD cells.

Consequently, we performed a new RNA-seq experiment analyzing RD and JR1 shSKP2.2 and shSCR cells to identify common transcriptional changes at the same time point.

The results are reported in Fig. S8.

Analysis of MYOD peaks on the *SKP2* region shows that MYOD deposition does not seem to be significantly modulated (Ancillary Figure 11 for Reviewers). This could be due to the specific time point but it is also in line with the modest modulation in term of global increase of MYOD peaks after SKP2 depletion (Fig. S8e and Results section).

It is, indeed, evident by our analysis that MYOD deposition increased after SKP2 knockdown at peaks of MYOD-target myogenic genes that are commonly up-regulated in the two cell lines, such as *CDKN1A*, *MYL1*, *MYOG* and *MYBPH*. Therefore, MYOD binding/transcriptional activity was induced in a SKP2-depleted context particularly on selected genes such as myogenic genes (Fig. S8g).

Overall, the induction of differentiation suggests that the stemness signature should be compromised. In accordance, analyzing the RNA-seq we discovered that several genes that are included in a stemness gene set (STEMNESS-UP) were down-regulated (see the Results section, Figs. S8a,b and Supplementary Table 2). Then, we analyzed MYOD binding on putative MYOD-bound gene regions previously identified by Tenente et al. (Ref. 7 Revised Manuscript) and included in the STEMNESS-UP gene set. However, we were unable to find any modulation of MYOD binding even on those stemness genes that were down-regulated after SKP2 knockdown suggesting that MYOD functions on these genes are not relevant in the response to SKP2 depletion.

3. The authors used β -gal staining assay to show induction of senescence upon SKP2 depletion. Could the authors validate the same on the molecular levels showing the levels of any senescent marker genes for example SASP protein either by quantitative real time PCR from the available cDNA or by western blot.

When we analyzed the new RNA-seq data on RD and JR1 cells we noticed a down-regulation of a subset of SASP genes (88 out of 112 total genes included in the gene set https://www.gsea-msigdb.org/gsea/msigdb/cards/REACTOME_SENESCENCE_ASSOCIATED_SECRETORY_PHENOTYPE_SASP), which have been reported in the Ancillary Table 1 for Reviewers.

Therefore, it appears conceivable that in our contest, senescence could be mainly related to the up-regulation of protein levels of p21, p27 and p57, which are pro-senescent factors (Refs. 45,50 Revised Manuscript and ¹³⁻¹⁵).

Moreover, it has been demonstrated that p21-induced senescence is often not associated to SASP response (Ref. 51 Revised Manuscript).

We added a couple of lines on p21 in the Discussion.

4. The authors have shown effect on cell proliferation upon SKP2 depletion in cell lines and the same effect is also seen in the reduced tumor volumes after SKP2 is silenced by shRNA or using the NAE inhibitor LN4924. Since the IHC sections have been made and stained for the relevant targets but it was surprising that the authors haven't stained for the proliferation marker Ki67 in these sections. The Ki67 staining for the IHC sections shown in Fig 6k and 7c would make it clear at the molecular level.

The staining for Ki67 has been done and added to the revised manuscript as requested.

To All Reviewers:

As reported in the Cover letter, we have replaced all the radiograms for Fig. S4d (previously Fig. S3d) of RD cells. Indeed, doing the uncropped figures, we noticed that the radiogram of p27^{Kip1} was erroneously duplicated from that of JR1 cells. Thus, we replaced all the radiograms for the investigated proteins in RD cells. We apologize for this inaccuracy.

References

1. Tintignac, L. A. *et al.* Degradation of MyoD Mediated by the SCF (MAFbx) Ubiquitin Ligase. *J. Biol. Chem.* **280**, 2847–2856 (2005).
2. Imaki, H. *et al.* Cell cycle-dependent regulation of the Skp2 promoter by GA-binding protein. *Cancer Res.* **63**, 4607–13 (2003).
3. Hume, S. *et al.* The NUCKS1-SKP2-p21/p27 axis controls S phase entry. *Nat. Commun.* **12**, 6959 (2021).
4. Wang, S., Raven, J. F. & Koromilas, A. E. STAT1 represses Skp2 gene transcription to promote p27^{Kip1} stabilization in Ras-transformed cells. *Mol. Cancer Res.* **8**, 798–805 (2010).
5. Lu, D. Y. *et al.* The ETS transcription factor ETV6 constrains the transcriptional activity of EWS–FLI to promote Ewing sarcoma. *Nat. Cell Biol.* (2023) doi:10.1038/s41556-022-01059-8.
6. Garraway, L. A. & Sellers, W. R. Lineage dependency and lineage-survival oncogenes in human cancer. *Nat. Rev. Cancer* **6**, 593–602 (2006).
7. Pomella, S. *et al.* Genomic and Epigenetic Changes Drive Aberrant Skeletal Muscle Differentiation in Rhabdomyosarcoma. *Cancers (Basel)*. **15**, 2823 (2023).
8. Singh, K. & Dilworth, F. J. Differential modulation of cell cycle progression distinguishes members of the myogenic regulatory factor family of transcription factors. *FEBS J.* **280**, 3991–4003 (2013).
9. Zhang, L. *et al.* FBXO22 promotes the development of hepatocellular carcinoma by regulating the ubiquitination and degradation of p21. *J. Exp. Clin. Cancer Res.* **38**, 101

- (2019).
10. Abbas, T. *et al.* PCNA-dependent regulation of p21 ubiquitylation and degradation via the CRL4^{Cdt2} ubiquitin ligase complex. *Genes Dev.* **22**, 2496–2506 (2008).
 11. Amador, V., Ge, S., Santamaría, P. G., Guardavaccaro, D. & Pagano, M. APC/CCdc20 Controls the Ubiquitin-Mediated Degradation of p21 in Prometaphase. *Mol. Cell* **27**, 462–473 (2007).
 12. Wang, F. *et al.* Ubiquitination of p21 by E3 Ligase TRIM21 Promotes the Proliferation of Human Neuroblastoma Cells. *NeuroMolecular Med.* **23**, 549–560 (2021).
 13. Majumder, P. K. *et al.* A Prostatic Intraepithelial Neoplasia-Dependent p27Kip1 Checkpoint Induces Senescence and Inhibits Cell Proliferation and Cancer Progression. *Cancer Cell* **14**, 146–155 (2008).
 14. Kovach, A. R. *et al.* Identification and targeting of a <scp>HES1-YAP1-CDKN1C</scp> functional interaction in fusion-negative rhabdomyosarcoma. *Mol. Oncol.* **16**, 3587–3605 (2022).
 15. Hernandez-Segura, A. *et al.* Unmasking Transcriptional Heterogeneity in Senescent Cells. *Curr. Biol.* **27**, 2652-2660.e4 (2017).

Ancillary Figure 1

Ancillary Figure 1. *SKP2* is regulated by a MYOD-bound enhancer.

a, Representative western blot (n = 3 independent experiments) of the indicated proteins on JR1 and RH30 cells transfected with either Scrambled (siSCR) or two different MYOD siRNA sequences (siMYOD.2 and siMYOD.3) at 24 hours (h) and 48h post-transfection. Vinculin is the loading control. **b**, mRNA levels (RT-qPCR) of *MYOD1* and *SKP2* on cells treated as in (a) were normalized to *GAPDH* levels and expressed as fold increase over siSCR. n = 3 independent experiments, data presented as mean values \pm SD, two-way ANOVA.

Ancillary Figure 2

Ancillary Figure 2. SKP2 depletion induces $p27^{Kip1}$ -dependent cell cycle arrest reducing growth.

a, Representative western blot (n = 3 independent experiments) of p57^{Kip2} on RD and JR1 cells transfected with either SCR, SKP2, p27^{Kip1} or SKP2 + p27^{Kip1} siRNA at 72h post-transfection. Vinculin is the loading control.

Ancillary Figure 3

Ancillary Figure 3. NEDDylation inhibition prevents SKP2 functions and induces G2/M cell cycle arrest.

a, Histogram depicts fold changes of the percentage of MLN4924 treated cells (60 nM for 24 hours) in G1, S and G2/M cell cycle phases over DMSO cells (1 arbitrary unit, not reported). $n = 3$ independent experiments, data presented as mean values \pm SD, Student's two-tailed t-test.

Ancillary Figure 4

Ancillary Figure 4. MYOD is not a direct substrate of SKP2.

a, Representative western blot (n = 3 independent experiments) of co-Immunoprecipitation of either endogenous SKP2 (left) or MYOD (right) in RD and JR1 cells showing SKP2 and MYOD.

Ancillary Figure 5

Ancillary Figure 5. E2F1 does not regulates *SKP2* expression in rhabdomyosarcoma cells.

a, Representative western blot ($n = 3$ independent experiments) of the indicated proteins on RD, JR1, RH4 and RH30 cells transfected with either Scrambled (siSCR) or two different E2F1 siRNA sequences (E2F1.1: UAACUGCACUUUCGGCCUUU and E2F1.2: CUACUCAGCCUGGAGCAAGAA, Sigma-Aldrich, St Louis, MO, USA) at 24 hours (h) post-transfection. Vinculin is the loading control. **b**, mRNA levels (RT-qPCR) of *E2F1* and *SKP2* on cells treated as in (a) were normalized to *GAPDH* levels and expressed as fold increase over siSCR. $n = 3$ independent experiments, data presented as mean values \pm SD, two-way ANOVA.

Ancillary Figure 6

Ancillary Figure 6. *MYOD1* is a strong and selective dependency in rhabdomyosarcoma cells.

a, Volcano plot identifying selective gene dependencies in rhabdomyosarcoma cells (effect size) among 17451 genes analyzed. **b**, Violin plot depicting *MYOD1* gene effect comparison between RMS cells and all other cell types present in DepMap public 23Q2. Data were obtained from DepMap (<https://depmap.org/portal/>).

Ancillary Figure 7

Ancillary Figure 7. MYOD depletion induces growth arrest and hampers myogenic differentiation.

a, Growth curve analysis of RD and JR1 cells transfected with either Scrambled (siSCR) or MYOD siRNA (siMYOD.1). $n = 3$ independent experiments, data presented as mean values \pm SD, Student's two-tailed t-test. **b**, Histogram depicts fold changes of the percentage of cells treated as in (a) in G1, S and G2/M cell cycle phases over siSCR cells (1 arbitrary unit, not reported). $n = 3$ independent experiments, data presented as mean values \pm SD, Student's two-tailed t-test. P-value $* \leq 0.05$, $** \leq 0.01$, $**** \leq 0.0001$. **c**, Representative western blot ($n = 2$ independent experiments) of the indicated proteins on RD and JR1 cells transfected with either Scrambled (siSCR) or MYOD siRNA (siMYOD.1) at 48 hours (h) post-transfection. Vinculin is the loading control. **d**, mRNA levels (RT-qPCR) of the reported genes on cells treated as in (c) were normalized to *GAPDH* levels and expressed as fold increase over siSCR. $n = 2$ independent experiments, data presented as mean values \pm SD, Student's two-tailed t-test. **e**, Representative immunofluorescence of RD and JR1 cells treated as in (c) stained for Myosin Heavy Chain (MyHC) (green). Nuclei were stained with DAPI (blue). Scale Bar = 50 μ m.

Ancillary Figure 8

Ancillary Figure 8. MYOD overexpression does not affect rhabdomyosarcoma cell growth.

a, Growth curve analysis of RD and JR1 infected with a Doxycycline-inducible lentiviral vector overexpressing human *MYOD1* (pINDUCER20-FLAG-hMYOD (#800), Addgene) and treated or not with 1 μg/mL of Doxycycline. n = 3 independent experiments, data presented as mean values ± SD, Student's two-tailed t-test. **b**, mRNA levels (RT-qPCR) of the reported genes on cells treated as in (a) for 72h were normalized to *GAPDH* levels and expressed as fold increase over Doxycycline-untreated cells. n = 3 independent experiments, data presented as mean values ± SD, Student's two-tailed t-test. **c**, Representative western blot (n = 3 independent experiments) of the indicated proteins on RD and JR1 cells treated as in (b). Vinculin is the loading control.

Ancillary Figure 9

a —●— siSCR —■— siSKP2
—▲— siMYOD.1 —▼— siSKP2+siMYOD.1

b ■ G1 ■ S □ G2/M

c

d

e

Ancillary Figure 9. MYOD-SKP2 co-silencing enhances cell growth arrest but impedes myogenic differentiation.

a, Growth curve analysis of RD and JR1 cells transfected with either Scrambled (SCR), SKP2, MYOD or SKP2 + MYOD siRNAs. $n = 3$ independent experiments, data presented as mean values \pm SD, Student's two-tailed t-test. **b**, Histogram depicts fold changes of the percentage of cells treated as in **(a)** in G1, S and G2/M cell cycle phases over siSCR cells (1 arbitrary unit, not reported). $n = 3$ independent experiments, data presented as mean values \pm SD, two-way ANOVA. P-value vs siSCR * ≤ 0.05 , ** ≤ 0.01 , *** ≤ 0.001 , **** ≤ 0.0001 ; P-value vs siSKP2 \$ ≤ 0.05 , \$\$ ≤ 0.01 , \$\$\$ ≤ 0.001 . **c**, Representative western blot ($n = 3$ independent experiments) of the indicated proteins on RD and JR1 cells treated as in **(a)**. Vinculin is the loading control. **d**, mRNA levels (RT-qPCR) of the reported genes on cells treated as in **(a)** for 72h were normalized to *GAPDH* levels and expressed as fold increase over siSCR. $n = 2$ independent experiments, data presented as mean values \pm SD. **e**, Representative immunofluorescence of RD and JR1 cells treated as in **(a)** stained for Myosin Heavy Chain (MyHC) (green). Nuclei were stained with DAPI (blue). Scale Bar = 50 μ m.

Ancillary Figure 10

Ancillary Figure 10. Sensitivity to MLN4924 correlates with *SKP2* expression but not with *MYOD1* expression

Pearson correlation analysis between MLN4924 AUC and (a) *MYOD1* expression or (b) *SKP2* expression performed on 721 cancer cell lines. Data were obtained from DepMap portal (<https://depmap.org/portal/>) using Expression Public 23Q2 dataset for gene expression data and Drug Sensitivity AUC (CTD²) dataset for MLN4924 AUC. TPM, Transcripts per Million; AUC, area under the curve.

a

Ancillary Figure 11. MYOD binding at *SKP2* locus after *SKP2* silencing

a, Profile of ChIP-seq read densities of MYOD (green), and RNA-seq (grey) at *SKP2* locus on RD cells.

Ancillary Table 1. RD and JR1 Commonly downregulated genes in REACTOME_SENESCENCE_ASSOCIATED_SECRETORY_PHENOTYPE_SASP gene set

GeneID	RD				JR1			
	shSCR	shSKP2	FC	LOG2(FC)	shSCR	shSKP2	FC	LOG2(FC)
ANAPC1	16.7	8.62	0.5435	0.6262	11.82	7.27	0.6451	0.7182
ANAPC11	52.05	33.03	0.6415	0.7150	37.54	31.66	0.8474	0.8855
ANAPC15	16.59	6.9	0.4491	0.5352	9.43	4.14	0.4928	0.5780
ANAPC2	29.5	23.37	0.7990	0.8472	23.87	18.72	0.7929	0.8423
ANAPC5	52.79	40.32	0.7682	0.8223	51.69	41.38	0.8043	0.8515
ANAPC7	37.21	22.77	0.6221	0.6979	29.23	19.6	0.6814	0.7497
CCNA2	27.32	17.65	0.6585	0.7299	10.1	6.1	0.6396	0.7134
CDC23	13.67	11.01	0.8187	0.8629	13.62	12.82	0.9453	0.9600
CDC26	17.85	12.28	0.7045	0.7694	23.25	12.79	0.5687	0.6495
CDC27	22.07	17.23	0.7902	0.8401	15.29	13.52	0.8913	0.9194
CDK2	39.09	12.99	0.3490	0.4319	20.74	5.86	0.3155	0.3957
CDK4	133.91	102.68	0.7685	0.8225	121.52	116.45	0.9586	0.9698
CDKN1B	15.57	11.06	0.7278	0.7890	11.15	9.53	0.8667	0.9005
CDKN2A	34.1	28.79	0.8487	0.8865	96.8	71.93	0.7457	0.8038
CDKN2B	2.79	2.38	0.8918	0.9198	10.13	7.97	0.8059	0.8527
CDKN2C	49.08	23.69	0.4930	0.5782	15.86	9.79	0.6400	0.7137
CXCL8	2.17	1.33	0.7350	0.7949	4.89	1.03	0.3447	0.4272
EHMT1	42.27	36.68	0.8708	0.9037	32.85	30.86	0.9412	0.9570
EHMT2	58.25	32.07	0.5581	0.6398	16.86	14.13	0.8471	0.8853
FOS	93.54	24.38	0.2685	0.3431	109.8	24.11	0.2266	0.2947
FZR1	29.2	18.93	0.6599	0.7311	20	12.44	0.6400	0.7137
H2AC14	411.76	100.48	0.2459	0.3171	112.08	26.52	0.2434	0.3143
H2AC18	2469.16	1019.57	0.4132	0.4989	892.71	359.27	0.4031	0.4886
H2AC20	647.78	226.18	0.3502	0.4331	234.35	113.3	0.4857	0.5711
H2AC4	120.82	37.23	0.3138	0.3938	41.97	11.99	0.3023	0.3811
H2AC6	318.32	154.69	0.4876	0.5730	127.23	84.43	0.6662	0.7366
H2AC7	91.2	40.08	0.4456	0.5316	31.65	12.73	0.4205	0.5064
H2AC8	260.48	90.85	0.3513	0.4343	94.02	35.98	0.3892	0.4742
H2AJ	13.31	11.82	0.8959	0.9229	21.37	9.31	0.4609	0.5468
H2AX	132.82	103.32	0.7796	0.8315	70.37	42.58	0.6106	0.6876
H2AZ2	48	38.4	0.8041	0.8513	27.5	25.64	0.9347	0.9521
H2BC1	257.59	65.05	0.2554	0.3282	64.67	26.64	0.4209	0.5068
H2BC10	295.63	81.15	0.2769	0.3527	91.21	18.33	0.2096	0.2746
H2BC11	242.87	84.88	0.3522	0.4353	76.4	30.61	0.4084	0.4941
H2BC12	894.47	351.33	0.3935	0.4787	381.36	167.23	0.4400	0.5260
H2BC13	189.63	39.15	0.2106	0.2757	78.21	15.95	0.2140	0.2798
H2BC14	231.47	51.01	0.2237	0.2913	58.26	12.53	0.2283	0.2967
H2BC15	216.79	111.62	0.5171	0.6013	93.26	47.01	0.5093	0.5939
H2BC17	363.76	132.77	0.3667	0.4507	125.05	45.8	0.3713	0.4555
H2BC21	133.8	78.91	0.5928	0.6716	67.33	62.75	0.9330	0.9508
H2BC3	114.28	47.77	0.4231	0.5090	38.04	17.11	0.4639	0.5498
H2BC4	647.34	279.74	0.4330	0.5191	215.06	175.05	0.8148	0.8598
H2BC5	415.74	164.12	0.3962	0.4815	103.84	79.16	0.7646	0.8193
H2BC6	213.22	47.88	0.2282	0.2965	103.93	20.41	0.2040	0.2679
H2BC7	234.27	69.24	0.2986	0.3769	75.52	24.29	0.3305	0.4120

H2BC8	219.58	98.25	0.4500	0.5360	90.2	41.73	0.4685	0.5544
H2BC9	346.12	111.08	0.3229	0.4037	92.94	28.43	0.3133	0.3932
H3-3A	96.12	78.88	0.8225	0.8659	68.95	66.48	0.9647	0.9743
H3-3B	111.46	79.71	0.7177	0.7805	98.81	74.54	0.7568	0.8130
H3C1	183.04	52.31	0.2897	0.3670	61.72	13.99	0.2390	0.3092
H3C10	528.48	111.75	0.2129	0.2785	156.37	31.33	0.2054	0.2696
H3C11	157.88	41.4	0.2669	0.3413	47.48	11.98	0.2677	0.3423
H3C12	212.49	68.85	0.3272	0.4084	69.73	15.61	0.2348	0.3043
H3C13	490.78	130.97	0.2684	0.3430	120.42	20.37	0.1760	0.2339
H3C14	11.37	8.19	0.7429	0.8015	16.64	15.95	0.9609	0.9715
H3C15	1746.3	641.32	0.3676	0.4517	402.3	116.32	0.2909	0.3684
H3C2	1039.78	316.65	0.3052	0.3843	264.5	92.55	0.3524	0.4355
H3C3	496.2	187.23	0.3786	0.4632	162.92	54.07	0.3360	0.4179
H3C4	472.54	124.46	0.2649	0.3391	120.96	35.84	0.3021	0.3808
H3C6	447.63	157.8	0.3540	0.4372	125.47	36.76	0.2986	0.3769
H3C7	495.91	161.94	0.3279	0.4092	153.47	43.37	0.2872	0.3643
H3C8	257.05	115.94	0.4532	0.5392	104.02	35.8	0.3504	0.4334
H4C1	261.09	67.46	0.2612	0.3348	71.9	14.45	0.2119	0.2773
H4C11	430.57	182	0.4240	0.5100	152.39	71.86	0.4750	0.5607
H4C12	485.91	207.27	0.4277	0.5137	271.17	88.44	0.3286	0.4099
H4C13	71.04	20.07	0.2925	0.3701	14.27	2.98	0.2606	0.3342
H4C14	1163.49	295.87	0.2549	0.3276	335.38	53.94	0.1633	0.2183
H4C15	1173.55	316.89	0.2706	0.3456	351.01	75.46	0.2172	0.2836
H4C2	470.85	154.03	0.3286	0.4099	145.8	57.2	0.3965	0.4818
H4C3	456.98	168.46	0.3700	0.4542	115.4	59.05	0.5159	0.6002
H4C4	549.27	101.84	0.1869	0.2472	149.7	28.12	0.1932	0.2549
H4C5	570.74	221.96	0.3900	0.4751	202.62	97.44	0.4834	0.5690
H4C8	596.03	260.4	0.4378	0.5239	188.98	98	0.5211	0.6051
H4C9	223.01	62.11	0.2817	0.3581	57.43	13.28	0.2444	0.3154
IL6	7.28	0.36	0.1643	0.2194	8.18	0.55	0.1688	0.2251
MAPK3	22.73	11.34	0.5200	0.6041	18.85	11.75	0.6423	0.7157
MAPK7	6.86	5.74	0.8575	0.8934	3.75	3.31	0.9074	0.9316
NFKB1	14.56	10.5	0.7391	0.7983	12.4	9.88	0.8119	0.8575
RELA	25.96	21.98	0.8524	0.8894	23.1	19.3	0.8423	0.8815
RPS27A	292.54	240.57	0.8230	0.8663	297.21	232.06	0.7815	0.8331
RPS6KA1	4.37	2.52	0.6555	0.7273	3.07	0.61	0.3956	0.4809
RPS6KA2	26.39	17.16	0.6630	0.7338	4.48	4.25	0.9580	0.9694
STAT3	43.14	34.14	0.7961	0.8449	51.76	35.24	0.6869	0.7544
UBB	774.12	567.77	0.7338	0.7939	691.64	477.24	0.6905	0.7574
UBC	456.28	377.65	0.8280	0.8703	475.52	457.55	0.9623	0.9725
UBE2C	208.12	149.44	0.7194	0.7819	78.7	44.51	0.5710	0.6517
UBE2E1	26.7	17.61	0.6718	0.7414	20.92	13.45	0.6592	0.7305
UBE2S	108.15	94.12	0.8715	0.9042	59.7	51.58	0.8662	0.9001

REVIEWERS' COMMENTS

Reviewer #1 (Remarks to the Author):

The authors have addressed most of the raised concerns.

Reviewer #2 (Remarks to the Author):

I have completed the review of the revised manuscript, and I am pleased to report that the authors have successfully addressed all the concerns and suggestions raised during the initial review process. I find the paper to be of high quality and believe it is now suitable for publication.

I have no further issues or recommendations to add at this time.

Thank you for considering my assessment, and I look forward to seeing this valuable contribution in print.

Reviewer #4 (Remarks to the Author):

I am satisfied with the authors explanations and recommend this for publication.

Reviewer #5 (Remarks to the Author):

I was asked to assess whether the authors had sufficiently addressed the concerns raised by reviewer #3. To address these, the authors provided additional experimental data as well as elegant data mining approaches. They furthermore adequately rephrased the passages concerning MYOD being pro-tumorigenic. Overall, I can confidently say that the authors have addressed the comments by reviewer #3 thoroughly.